# Benchmarking Physics-Informed Time-Series Models for Operational Global Station Weather Forecasting

**Tao Han** [1] [2]  **Zhibin Wen** [3]  **Zhenghao Chen** [4]  **Dazhao Du** [5]  **Song Guo** [1]  **Lei Bai** [2]

## Abstract

The development of Time-Series Forecasting (TSF) models is often constrained by the lack of comprehensive datasets, especially in Global Station Weather Forecasting (GSWF), where existing datasets are small, temporally short, and spatially sparse. To address this, we introduce WEATHER-5K, a large-scale observational weather dataset that better reflects real-world conditions and supports improved model training and evaluation. While recent TSF methods perform well on benchmarks, they still lag behind operational Numerical Weather Prediction (NWP) systems in capturing complex weather dynamics and extreme events. We propose PhysicsFormer, a physics-informed forecasting model that combines a dynamic core with a Transformer residual to predict future weather states. Physical consistency is enforced via pressure–wind alignment and energy-aware smoothness losses, ensuring plausible dynamics while capturing complex temporal patterns. We benchmark PhysicsFormer and other TSF models against operational systems across several weather variables, extreme event prediction, and model complexity, providing a comprehensive assessment of the gap between academic TSF models and operational forecasting. The dataset and benchmark implementation are available at: https://github.com/taohan10200/WEATHER-5K.

[1]Department of Computer Science and Engineering, Hong Kong University of Science and Technology, Hong Kong SAR China [2]Shanghai Artificial Intelligence Laboratory, Shanghai, China [3]Department of Computer Science and Engineering, Southern University of Science and Technology, Shenzhen, China [4]School of Computer and Information Sciences, University of Newcastle, Newcastle, Australia [5]Hangzhou Innovation Institute of Beihang University, Hangzhou, China. Correspondence to: Song Guo <songguo@ust.hk>, Lei Bai <bailei@pjlab.org.cn>.

*Proceedings of the 43rd International Conference on Machine Learning*, Seoul, South Korea. PMLR 306, 2026. Copyright 2026 by the author(s).

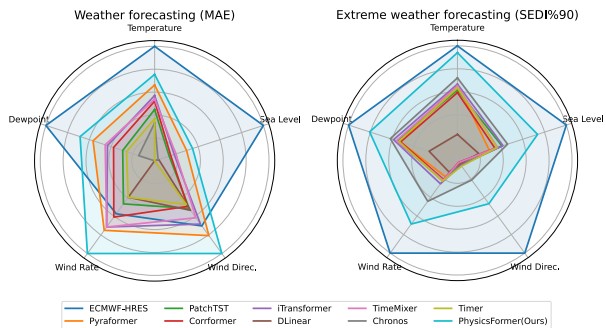

*Figure 1.* Comparison of the operational NWP model and various TSF models across five weather variables. On the left-hand side, the prediction comparison is based on the MAE metric, and on the right-hand side, it focuses on predictions under extreme weather conditions. In both cases, a larger area indicates better performance. MAE values are normalized and inverted, so larger unitless scores indicate better performance.

## 1. Introduction

Global Station Weather Forecasting (GSWF) is essential for providing precise and timely weather information, with significant implications across various sectors. The mainstream approach for precise station-based weather forecasting has traditionally been Numerical Weather Prediction (NWP), which refers to physics-based methods that solve partial differential equations (Phillips, 1956; Lynch, 2008). Despite their reliability, physically-based NWP models remain computationally intensive, requiring substantial resources for data assimilation, medium-range forecasting, and interpolation to produce station-specific predictions. Recently, data-driven weather forecasting models (Lam et al., 2023; Bi et al., 2023; Chen et al., 2025; Han et al., 2024) have emerged as a complementary paradigm, leveraging historical weather data and advanced machine learning techniques to improve forecasting accuracy and efficiency.

Time-Series Forecasting (TSF) methods provide a lightweight alternative by directly learning from historical station observations, enabling end-to-end predictions without complex interpolation. TSF approaches have recently shown strong performance on small-scale station datasets; for example, (Zhou et al., 2022; Zeng et al., 2023; Liu et al., 2024a; Wang et al., 2024; Murad et al., 2025) achieve

high accuracy in temperature forecasting on the Weather dataset[1]. These results highlight the potential of TSF as a computationally efficient and scalable alternative to NWP for station-level forecasting. In this work, we systematically evaluate this potential through a comprehensive benchmark.

In addition, to combine the efficiency of TSF with physical consistency, physics-informed methods (Sel et al., 2023; Nagda et al., 2025; Nasiri et al., 2026) have been explored. Building on this, we propose PhysicsFormer, which integrates a dynamic core into a Transformer backbone and enforces physics constraints via the loss function, aiming to capture complex dynamics and extreme events while remaining computationally efficient.

### 1.1. Knowledge Gap and Motivations

We first identify four major limitations (**L1–L4**) as follows:

**L1: Inadequate Data Scale and Diversity for Global Generalization.** Current TSF methods for GSWF are limited by small, homogeneous datasets. Many datasets focus on a single station (Wu et al., 2021), a localized region (Godahewa et al., 2021), or short timeframes (Wu et al., 2023), which restricts models' ability to generalize. As a result, these models typically excel only in short-term forecasts, such as one-day predictions. Even though recent work has introduced end-to-end GSWF models (Wu et al., 2023), the lack of comprehensive datasets remains a major bottleneck.

**L2: Disconnect Between Academic TSF Models and Real-world Forecasting.** Though state-of-the-art TSF methods excel under ideal conditions (e.g., short lead times) on academic benchmarks, their real-world deployment remains limited. This gap highlights the need for an enhanced benchmark that aligns with operational challenges, captures the complexities of real-world weather systems (Wu et al., 2021; Zhou et al., 2021), and bridges the divide between academic research and practical application.

**L3: Insufficiently Operational Evaluation Metrics.** The evaluation of TSF methods for GSWF often relies on limited metrics such as Mean Squared Error (MSE) (Wu et al., 2021; Nie et al., 2023; Liu et al., 2022), which focus solely on overall accuracy. This overlooks key aspects of weather forecasting, including the prediction of extreme events such as heatwaves, storms, and strong winds, which are vital for real-world applications. Moreover, factors such as model complexity are usually not considered, resulting in an incomplete assessment of real-world forecasting performance.

**L4: Ignoring Physical Dynamics in Current TSF Methods.** Existing TSF methods (Wang et al., 2024; Murad et al., 2025) often ignore physical dynamics, such as pressure–wind relationships and energy conservation. This lim-

its forecasting accuracy and extreme event prediction.

### 1.2. Contribution and Innovation

Motivated by these limitations, we introduce WEATHER-5K, a new benchmark enabling operational TSF models, along with an enhanced evaluation framework using operational metrics for fair assessment. We further propose PhysicsFormer, a physics-informed TSF model. Our key contributions (**C1–C4**) are summarized below:

**C1: A Comprehensive Global Dataset, WEATHER-5K.** WEATHER-5K addresses L1 by providing extensive training data from *5,672* weather stations worldwide, spanning *10 years* of hourly observations. Its broad coverage and diverse conditions allow TSF methods to generalize across regions and time scales.

**C2: Enhanced Operational Benchmarking with NWP Comparisons.** To address L2, our benchmark evaluates a wide range of TSF models, including Transformer-, MLP-, GNN-based methods, and LTMs, against operational NWP models over longer lead times and diverse real-world scenarios, extending beyond prior work (Wu et al., 2023).

**C3: Operational Evaluation Metrics.** To address L3, we introduce the *Symmetric Extremal Dependence Index (SEDI)* for extreme event prediction and incorporate a complexity metric to assess efficiency and operational feasibility, providing a comprehensive operational evaluation.

**C4: Physics-Informed Forecasting.** To address L4, we propose PhysicsFormer, a physics-informed Transformer that integrates a dynamic core and enforces physics constraints via the loss function, improving real-world applicability and forecast accuracy.

## 2. Taxonomy of TSF Methods

We classify recent TSF models into four types: (a) MLP-based models, (b) Transformer-based models, (c) GNN-based models, and (d) large time-series models (LTMs) capable of zero-shot prediction.

**MLP-based models.** Many MLP-based forecasting models have been proposed due to their architectural simplicity (Oreshkin et al., 2020; Challu et al., 2023). STID (Shao et al., 2022) shows that a simple MLP can outperform sophisticated GNN models in spatio-temporal forecasting, while DLinear (Zeng et al., 2023) demonstrates that linear projections may surpass Transformer-based models. SparseTSF (Lin et al., 2024) exploits Cross-Period Sparse Forecasting to improve efficiency, while TimeMixer (Wang et al., 2024) and WPMixer (Murad et al., 2025) capture multi-resolution temporal information. Due to their high computational efficiency, these MLP-based models are particularly suitable for edge-side deployment.

---

[1] https://www.bgc-jena.mpg.de/wetter

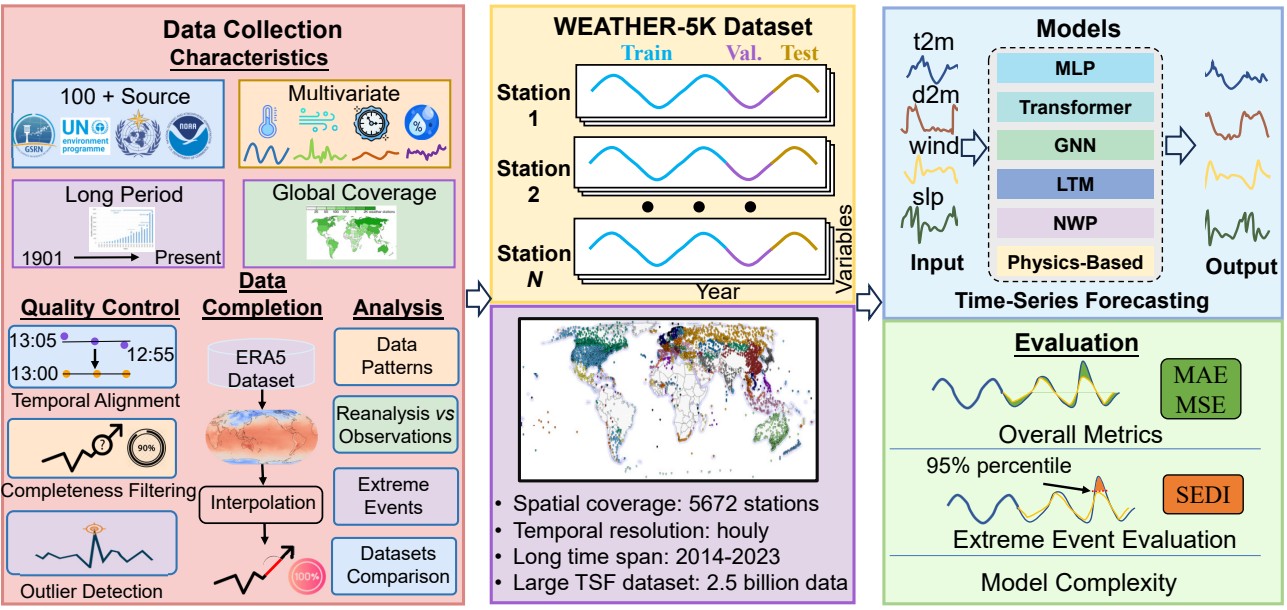

*Figure 2.* The flow diagram of the WEATHER-5K dataset and the benchmark.

**Transformer-based models.** The Transformer architecture (Vaswani et al., 2017) has become a prevalent backbone for TSF, giving rise to numerous variants. Informer (Zhou et al., 2021), Autoformer (Wu et al., 2021), Pyraformer (Liu et al., 2022), and FEDformer (Zhou et al., 2022) aim to improve efficiency for long-sequence modeling through sparse attention designs. PatchTST (Nie et al., 2023) enhances efficiency and performance by embedding time series as patches within a standard Transformer. iTransformer (Liu et al., 2024a) models inter-variable dependencies by treating each variable as a token. Corrformer (Wu et al., 2023) is tailored for worldwide weather station forecasting, explicitly capturing spatial and temporal correlations.

**GNN-based models.** Graph Neural Networks (GNNs) are effective for modeling interactions in multivariate time series by representing variables as nodes and relationships as edges. In spatio-temporal forecasting, GNNs are often combined with RNNs or TCNs, as exemplified by DCRNN (Li et al., 2018), AGCRN (Bai et al., 2020), and STGCN (Yu et al., 2018). To reduce reliance on predefined graph structures, MTGNN (Wu et al., 2020) and DGCRN (Li et al., 2023) learn adjacency matrices adaptively. Despite their strong performance in domains such as traffic forecasting, GNN-based models face scalability and computational challenges when applied to large-scale graphs.

**Large time series models (LTMs).** Motivated by the success of large language models (LLMs) in NLP (Raffel et al., 2020; Achiam et al., 2023), recent work explores their application to time-series forecasting or develops specialized large time series models (Liang et al., 2024). Some approaches directly adapt pre-trained LLMs for time se-

ries (Zhou et al., 2023; Jin et al., 2024; Xue & Salim, 2024; Liu et al., 2024b), but their zero-shot performance is often limited and typically requires task-specific fine-tuning. To address this, recent studies pre-train large time series models from scratch on large-scale time series datasets (Liu et al., 2024c; Goswami et al., 2024; Ansari et al., 2024). Representative examples include Chronos (Ansari et al., 2024), which adopts a T5-based (Raffel et al., 2020) language model architecture for tokenized time series, and Timer (Liu et al., 2024c), which trains an autoregressive Transformer on large-scale forecasting data. These models exhibit improved zero-shot forecasting performance.

## 3. WEATHER-5K Dataset

### 3.1. Collection and Pre-processing

**Data source.** WEATHER-5K is derived from global near-surface in-situ observations archived in the publicly available Integrated Surface Database (Smith et al., 2011), leveraging data from high-quality observation networks. ISD is a global repository of hourly and synoptic surface observations that encompasses a wide range of meteorological parameters, such as wind speed and direction, temperature, dew point, and sea level pressure.

**Station selection.** ISD contains records from over $20,000$ stations spanning 1901 to the present. However, certain stations are no longer operational, many do not report data on an hourly basis, and numerous stations have missing values for critical weather elements. To enhance data quality, a meticulous selection process was conducted to include only long-term, hourly reporting stations that are currently

*Table 1.* Statistics of different time-series datasets. 'N/A' means the dataset is not publicly available.

| Dataset | Domain | Frequency | Lengths | Stations | Variables | Year | Volume |
|---|---|---|---|---|---|---|---|
| Exchange (Lai et al., 2018) | Exchange | 1 day | 7,588 | 1 | 8 | 1990-2010 | 623KB |
| Electricity (Trindade, 2015) | Electricity | 1 hour | 26,304 | 321 | 1 | 2016-2019 | 92MB |
| ETTm2 (Zhou et al., 2021) | Electricity | 15 mins | 57,600 | 1 | 7 | 2016-2018 | 9.3MB |
| Traffic (Wu et al., 2021) | Traffic | 1 hour | 17,544 | 862 | 1 | 2016-2018 | 131MB |
| LargeST-CA (Liu et al., 2023) | Traffic | 5 mins | 525,888 | 8600 | 1 | 2017-2021 | 36.8GB |
| Solar (Lai et al., 2018) | Weather | 10 mins | 52,560 | 137 | 1 | 2006 | 8.3MB |
| Wind (Li et al., 2022) | Weather | 15 mins | 48,673 | 1 | 7 | 2020-2021 | 2.7MB |
| Weather (Wu et al., 2021) | Weather | 10 mins | 52,696 | 1 | 21 | 2020 | 7.0MB |
| Weather-Australia (Godahewa et al., 2021) | Weather | 1 day | 1,332∼65,981 | 3,010 | 4 | unknown | 202MB |
| GlobalTempWind (Wu et al., 2023) | Weather | 1 hour | 17,544 | 3,850 | 2 | 2019-2020 | 1034MB |
| CMA_Wind (Wu et al., 2023) | Weather | 1 hour | 17,520 | 34,040 | 1 | 2018-2019 | N/A |
| WEATHER-5K (Ours) | Weather | 1 hour | 87,648 | 5672 | 5 | 2014-2023 | 40.0GB |

operational and provide essential observations. As a result, $10,701$ stations were identified as continuously operating from 2014 to 2024.

## 3.2. Quality Control

A high-quality dataset is crucial for TSF. As shown in Figure 2, WEATHER-5K has been subjected to rigorous post-processing and quality control to ensure data reliability.

**Data interpretation.** In ISD, meteorological variables are encoded as strings rather than floating-point values, with each string containing the numerical measurement along with metadata such as quality flags and reporting types. For example, a temperature entry $< +0130, 1 >$ denotes a temperature of $13.0°C$ at the first quality level. Following the official ISD guidelines, all variables are systematically parsed into standardized numerical formats suitable for analysis and storage.

**Temporal alignment.** Although selected stations nominally report hourly data, timestamps may deviate slightly from exact hours (e.g., 12:55 or 13:05 instead of 13:00). To address this issue, we estimate missing hourly values using the nearest observations within a 30-minute window, substantially improving hourly coverage. For the small fraction of hours still missing due to the absence of nearby observations, linear interpolation is applied using data from the surrounding 12 consecutive hours to maintain temporal consistency.

**Completeness filtering.** From the 10,701 candidate stations, we retain only those with more than 90% valid hourly records. This filtering yields 5,672 stations worldwide, covering the period from 2014 to 2023, and balances long-term operation, hourly data availability, and coverage of diverse meteorological variables.

**Outlier detection.** Outliers are identified through an examination of temporal dynamics, targeting observations that fall far outside plausible ranges or exhibit anomalous behavior relative to the majority of records. We employ a combination of statistical methods and machine learning algorithms to distinguish genuine extremes from noise or data errors.

## 3.3. Data Completion and Statistics

After quality control, only a small fraction of data points remain missing, primarily due to long-term gaps at certain stations (e.g., exceeding one day). Since complete station records are required for spatial modeling in TSF, we fill remaining gaps using the ERA5 reanalysis dataset (Copernicus Climate Change Service, 2023) at station locations. Given that ERA5 is a widely used high-quality global reanalysis product, this interpolation introduces minor errors, with the largest impact expected for high-frequency variables such as wind speed.

To support model training on WEATHER-5K, we compute the decadal mean and variance of each variable for input standardization (see Appendix Table 6). In addition, motivated by the limited evaluation of extremes in existing TSF benchmarks, we calculate station-wise percentiles at $90\%, 95\%, 98\%$, and $99.5\%$ for each variable, enabling the assessment of extreme-value forecasting performance.

## 3.4. Analysis

As shown in Figure 3, WEATHER-5K provides a more accurate representation of real-world weather conditions than ERA5. Specifically, Figure 3**a)** presents the daily temperature variations in Chongqing, China (station ID: 57516099999), from July to September 2022. During this heatwave period, both the average and maximum temperatures in WEATHER-5K exceed those of ERA5 on most heatwave days, suggesting that ERA5 underestimates the diurnal temperature range at this station. Furthermore, Figure 3**b)** reports the RMSE of ERA5 against in-situ observations (Jiao et al., 2021), highlighting the discrepancy between reanalysis data and real-world observations and underscoring the importance of constructing weather datasets from actual in-situ measurements. We provide additional analyses of

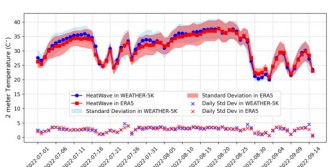
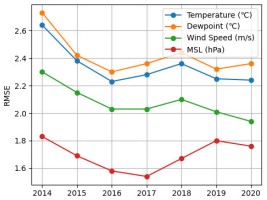

(a) Heatwave time series in WEATHER-5K    (b) The MSE between WEATHER-5K and ERA5

*Figure 3.* Comparison between WEATHER-5K and ERA5. (a) The daily 2m temperature at station 57516099999 from 1 July to 15 September 2022, where filled areas represent the variance from the daily mean. (b) MSE comparison between WEATHER-5K and ERA5 across different meteorological variables.

WEATHER-5K, including station density, different data patterns, and comparisons with ERA5, in Appendix D. Briefly, WEATHER-5K exhibits notable regional disparities in station coverage and heterogeneous temporal patterns across variables. In addition, Table 1 compares WEATHER-5K with widely used TSF datasets and highlights key limitations of existing benchmarks. *(1) Small scale and outdated.* Mainstream TSF datasets (Lai et al., 2018; Trindade, 2015; Wu et al., 2021) remain limited in scale and temporal coverage. For example, commonly used electricity consumption and exchange-rate datasets are sparse or outdated, constraining the practical applicability of forecasting models. *(2) Lagging behind other fields.* Compared to domains such as vision and language, where large-scale datasets (*e.g.*, Common Crawl and LAION-5B (Schuhmann et al., 2022)) have driven substantial scientific and economic progress, the TSF community has been slow to adopt large-scale data. Until recently, only one large-scale time-series dataset, LargeST (Liu et al., 2023), has been introduced. WEATHER-5K further addresses this gap by providing a large-scale, up-to-date observational weather dataset.

## 4. TSF Benchmarks on WEATHER-5K

### 4.1. Problem Definition

Consider $N$ weather stations worldwide, each collecting $V$ meteorological variables. The data from all weather stations can be represented as a spatiotemporal time series $X \in \mathbb{R}^{N \times T \times V}$ for a fixed look-back window of length $T$. At time $t$, the forecasting task is to predict $\hat{X}_{t+1:t+\tau} = \{X_{t+1}, ..., X_{t+\tau}\}$ based on the past $T$ frames $X_{t-T+1:t} = \{X_{t-T+1}, ..., X_t\}$. Here, $\tau$ is the length of the forecast horizon. Let $X$ and $\hat{X}$ denote the observed and forecasted data, respectively. The GSWF process can then be formulated as a mapping $\hat{X} = \mathcal{M}(X)$, where $\mathcal{M}$ represents a time-series forecasting method. For example, by setting $N = 1$ and ignoring spatial information, many state-of-the-art time-series forecasting methods (Li et al., 2022; Zhou et al., 2021; Liu et al., 2022; Wu et al., 2021; Zhou et al., 2022; Gu & Dao, 2024; Wang et al., 2024; Nie

et al., 2023; Liu et al., 2024a) can be applied to this task. When $N$ represents multiple spatially distributed stations, spatiotemporal methods (Wu et al., 2023; 2020; Shao et al., 2022) can also be applied.

### 4.2. Baselines

We compare 16 recent baselines, covering both temporal-only and spatiotemporal TSF models, categorized as follows:

**1) Transformer-based** methods, including popular transformer-based long-term forecasting models from 2021 to 2024: Informer (2021) (Zhou et al., 2021), Autoformer (2021) (Wu et al., 2021), Pyraformer (2022) (Liu et al., 2022), FEDformer (2022) (Zhou et al., 2022), PatchTST (2023) (Nie et al., 2023), iTransformer (2024) (Liu et al., 2024a), and Corrformer (2023) (Wu et al., 2023).

**2) MLP-based** models: STID (2022) (Shao et al., 2022), DLinear (2023) (Zeng et al., 2023), SparseTSF (2024) (Lin et al., 2024), TimeMixer (2024) (Wang et al., 2024), and WPMixer (2025) (Murad et al., 2025).

**3) GNN-based** model: MTGNN (2020) (Wu et al., 2020) is included to explore graph-based weather modeling.

**4) LTMs**: Large time series models such as Chronos (2024) (Ansari et al., 2024) and Timer (2024) (Liu et al., 2024c) are also evaluated on WEATHER-5K.

**5) Physics-based NWP**: The Numerical Weather Prediction model ECMWF-HRES (European Centre for Medium-Range Weather Forecasts, 2024), a leading global weather forecasting system, is included in WEATHER-5K for evaluation. It requires no training; its forecast products at corresponding time steps are downloaded and interpolated to the station locations in WEATHER-5K for assessment.

### 4.3. Physics-Informed Transformer

We propose PhysicsFormer, a hybrid spatiotemporal forecasting framework that combines a Transformer encoder–decoder with a graph-based neural dynamic core. Unlike grid-based numerical weather prediction models, our observations are collected from $N = 5{,}672$ highly irregularly distributed weather stations, making direct finite-difference computation of spatial gradients, such as $\nabla P$, infeasible. To address this challenge, PhysicsFormer first constructs a station-level spatial graph to approximate local physical derivatives and then integrates a lightweight neural ODE dynamic core to produce a physically guided base forecast. A Transformer residual branch further corrects unresolved local anomalies and model errors. The final prediction is therefore decomposed as

$$\hat{X}_{t+1:t+\tau} = X_{t+1:t+\tau}^{\text{phys}} + \Delta X_{t+1:t+\tau}^{\text{res}}, \qquad (1)$$

*Table 2.* Benchmarks on WEATHER-5K. The results are based on four prediction lengths: 24, 72, 120, and 168, with an input length of 48. For each method, we report the average results across all prediction lengths; full results are provided in Appendix Tables 8 and 9. ECMWF-HRES is the physics-based NWP model and represents the most accurate weather forecasting model. Underlining indicates the second-best performance.

| Type | Method | Temperature | | Dewpoint | | Wind Speed | | Sea Level Pressure | |
|---|---|---|---|---|---|---|---|---|---|
| | | MAE | MSE | MAE | MSE | MAE | MSE | MAE | MSE |
| NWP (Operational) | ECMWF-HRES | **1.94** | **8.56** | **2.06** | **9.96** | 1.56 | 5.00 | **1.29** | **5.17** |
| Transformer-based | Informer (2021 AAAI) | 2.75 | 15.20 | 2.87 | 17.98 | 1.52 | 4.88 | 4.17 | 40.63 |
| | Autoformer (2021 NIPS) | 2.82 | 16.44 | 2.98 | 18.41 | 1.55 | 5.20 | 4.17 | 41.46 |
| | Pyraformer (2022 ICLR) | 2.49 | 13.49 | 2.68 | 15.82 | 1.51 | 4.87 | 3.72 | 34.32 |
| | FEDformer (2022 ICML) | 2.85 | 16.84 | 2.98 | 19.07 | 1.57 | 5.20 | 4.03 | 37.82 |
| | PatchTST (2023 ICLR) | 2.84 | 17.18 | 3.07 | 20.47 | 1.59 | 5.35 | 4.27 | 44.40 |
| | Corrformer (2023 NMI) | 2.72 | 15.17 | 2.95 | 18.22 | 1.55 | 5.00 | 4.22 | 43.43 |
| | iTransformer (2024 ICLR) | 2.64 | 15.24 | 2.87 | 18.34 | 1.52 | 4.96 | 4.11 | 42.12 |
| MLP-based | STID (2022 CIKM) | 4.71 | 41.40 | 4.22 | 35.06 | 1.69 | 6.29 | 5.77 | 64.64 |
| | DLinear (2023 AAAI) | 3.57 | 23.71 | 3.49 | 23.38 | 1.61 | 5.32 | 4.60 | 46.28 |
| | SparseTSF (2024 ICML) | 3.24 | 20.53 | 3.15 | 21.43 | 1.66 | 5.80 | 4.98 | 54.70 |
| | TimeMixer (2024 ICLR) | 2.69 | 15.38 | 2.84 | 17.82 | 1.52 | 4.99 | 4.06 | 39.69 |
| | WPMixer (2025 AAAI) | 2.79 | 16.50 | 2.94 | 18.85 | 1.56 | 5.20 | 4.20 | 41.43 |
| GNN-based | MTGNN (2020 KDD) | 10.33 | 171.74 | 9.54 | 149.24 | 2.10 | 8.28 | 7.00 | 88.51 |
| LTMs | Chronos (2024 TMLR) | 3.03 | 20.76 | 3.28 | 24.57 | 1.72 | 6.64 | 4.62 | 53.58 |
| | Timer (2024 ICML) | 2.97 | 18.63 | 3.12 | 21.01 | 1.61 | 5.55 | 4.73 | 51.76 |
| Physics-based | PhysicsFormer (Ours) | 2.34 | 11.86 | 2.51 | 14.01 | **1.44** | **4.55** | 3.53 | 32.41 |

where $X^{\text{phys}}$ denotes the dynamic-core forecast and $\Delta X^{\text{res}}$ denotes the Transformer-based residual correction.

**Global Station Spatiotemporal Encoding.** Each weather station is associated with both dynamic observation data and a static geographical location. We compute an embedding for global station observation data $X_{t-T+1:t} \in \mathbb{R}^{B \times N \times T \times V}$:

$$E_{t-T+1:t} = \text{Emb}_{\text{var}}(X_{t-T+1:t}) + \text{Emb}_{\text{geo}}(\Lambda, \Phi) \\ + \text{Emb}_{\text{time}}(\overline{T}) \in \mathbb{R}^{B \times N \times D} \quad (2)$$

where $\text{Emb}_{var}(\cdot)$ encodes the historical sequences of $V$ variables by flattening the temporal and variable dimensions for each station and projecting them via an MLP. $\text{Emb}_{geo}(\cdot)$ encodes the static latitude and longitude $\Lambda, \Phi \in \mathbb{R}^{B \times N}$ using Fourier features followed by an MLP. $\text{Emb}_{time}(\cdot)$ captures temporal periodicity by averaging the look-back window time markers (e.g., month, day, weekday, hour) and applying Fourier feature encodings with an MLP projection.

**Graph-Based Dynamic Core.** To mimic simplified atmospheric dynamics over irregularly distributed stations, we construct a spatial graph $\mathcal{G} = (\mathcal{V}, \mathcal{E})$, where each node corresponds to a weather station and edges are built using $K$-nearest neighbors under the Haversine distance. This graph enables local message passing among geographically nearby stations and provides a discrete approximation to spatial derivatives such as pressure gradients, advection, and divergence.

For a meteorological variable $u$, the graph-based gradient at station $i$ is approximated by

$$\nabla u_i \approx \sum_{j \in \mathcal{N}(i)} \frac{w_{ij}}{\sum_{j \in \mathcal{N}(i)} w_{ij}} (u_j - u_i) W_\nabla, \quad (3)$$

where $\mathcal{N}(i)$ denotes the neighborhood of station $i$, $w_{ij}$ is a distance-based edge weight, and $W_\nabla$ is a learnable projection for gradient estimation.

Let $X_{i,t} = [T_{i,t}, W_{i,t}, P_{i,t}, D_{i,t}]$ denote the station state consisting of temperature, wind, pressure, and dew point. The dynamic core models the continuous-time evolution of each station state through a graph neural ODE:

$$\frac{dX_{i,t}}{dt} = \Phi_\theta (X_{i,t}, \nabla X_{i,t}), \quad (4)$$

where $\Phi_\theta$ is a lightweight GNN parameterizing coupled physical tendencies, including pressure-gradient-driven momentum changes, wind-induced thermodynamic advection, and mass-related divergence effects. Starting from the latest observed state, we autoregressively integrate the neural

*Table 3.* Extreme-event forecasting evaluation using SEDI (%) for the NWP model and TSF models. SEDI is calculated using 120-step predictions. The reported percentiles include both lower and upper percentile thresholds. Underlining indicates the second-best performance.

| Type | Methods | Temperature | | Dewpoint | | Wind Speed | | Wind Direction | | Sea Level Pressure | |
|---|---|---|---|---|---|---|---|---|---|---|---|
| | | 99.5th↑ | 90th↑ | 99.5th↑ | 90th↑ | 99.5th↑ | 90th↑ | 99.5th↑ | 90th↑ | 99.5th↑ | 90th↑ |
| NWP (Operational) | ECMWF-HRES | **37.4** | **82.6** | **35.4** | **76.4** | **10.2** | **40.8** | **13.1** | **45.4** | **77.5** | **89.7** |
| Transformer-based | Informer | 11.8 | 49.5 | 9.2 | 39.2 | 2.1 | 6.7 | 0.12 | 2.9 | 9.8 | 35.7 |
| | Autoformer | 12.4 | 52.1 | 8.3 | 38.9 | 0.3 | 7.8 | 0.13 | 1.6 | 10.4 | 32.1 |
| | Pyraformer | 10.7 | 54.8 | 7.2 | 40.1 | 0.6 | 7.2 | 0.06 | 1.1 | 10.5 | 26.2 |
| | FEDformer | 11.9 | 50.9 | 9.9 | 40.7 | 2.9 | 9.5 | 0.08 | 0.7 | 7.5 | 21.4 |
| | PatchTST | 10.9 | 50.8 | 8.9 | 42.4 | 0.5 | 8.9 | 0.10 | 2.2 | 13.5 | 36.7 |
| | Corrformer | 10.9 | 48.9 | 8.4 | 39.9 | 1.7 | 8.4 | 0.12 | 0.9 | 8.9 | 30.9 |
| | iTransformer | 14.1 | 55.0 | 10.4 | 44.8 | 1.3 | 10.3 | 0.14 | 2.3 | 15.9 | 37.5 |
| MLP-based | STID | 0.0 | 1.4 | 0.0 | 2.3 | 0.1 | 0.1 | 0.00 | 0.0 | 0.0 | 0.2 |
| | DLinear | 5.8 | 18.8 | 3.2 | 19.9 | 0.3 | 5.1 | 0.13 | 1.7 | 2.8 | 17.5 |
| | SparseTSF | 6.1 | 41.1 | 9.7 | 43.4 | 1.1 | 10.5 | 0.13 | 3.7 | 7.5 | 31.5 |
| | TimeMixer | 11.9 | 53.7 | 8.9 | 42.7 | 1.1 | 8.7 | 0.01 | 1.0 | 11.7 | 33.0 |
| | WPMixer | 11.9 | 53.1 | 8.2 | 42.3 | 0.8 | 8.4 | 0.01 | 1.0 | 10.6 | 33.1 |
| GNN-based | MTGNN | 3.1 | 9.9 | 2.9 | 9.0 | 0.1 | 4.0 | 0.08 | 0.5 | 1.1 | 3.9 |
| LTMs | Chronos | 26.2 | 59.5 | 16.9 | 47.1 | 6.3 | 18.0 | 2.51 | 9.6 | 18.7 | 41.3 |
| | Timer | 14.2 | 52.3 | 8.4 | 41.0 | 1.2 | 9.1 | 0.13 | 2.8 | 9.4 | 32.6 |
| Physics-based | PhysicsFormer (Ours) | 30.6 | 77.7 | 20.4 | 61.6 | 7.8 | 28.0 | 5.01 | 21.2 | 62.6 | 66.0 |

dynamics to obtain the physical forecast:

$$X_{t+k}^{\text{phys}} = X_{t+k-1}^{\text{phys}} + \int_{t+k-1}^{t+k} \Phi_\theta \left( X^{\text{phys}}(s), \mathcal{G} \right) ds, \quad (5)$$
$$k = 1, \dots, \tau.$$

In practice, the integral is approximated using a fourth-order Runge–Kutta solver, which bridges the continuous dynamic formulation with discrete multi-step forecasting.

**Spatiotemporal Interaction Modeling.** Given the global station embeddings $E_{t-T+1:t}$, we apply a Transformer encoder to model interactions among global stations:

$$H_{enc} = \text{Encoder}(E_{t-T+1:t}) \in \mathbb{R}^{B \times N \times D} \quad (6)$$

Through self-attention, the encoder enables global information flow across stations, capturing global spatial interactions and large-scale atmospheric dependencies. The decoder receives the recent historical observations of length $L$ and the encoder output to generate future predictions:

$$Z_{dec} = \text{Decoder}(E_{dec}, H_{enc}) \in \mathbb{R}^{B \times N \times D} \quad (7)$$

where $E_{dec}$ is embedded in the same way as the encoder input.

**Physics-Guided Residual Forecasting.** The decoder output is mapped to a residual correction over the dynamic-core trajectory:

$$\Delta X_{t+1:t+\tau}^{\text{res}} = \text{Proj}(Z_{dec}) \in \mathbb{R}^{B \times N \times \tau \times V}. \quad (8)$$

The final forecast is obtained by adding this residual correction to the physically guided base forecast:

$$\hat{X}_{t+1:t+\tau} = X_{t+1:t+\tau}^{\text{phys}} + \Delta X_{t+1:t+\tau}^{\text{res}}. \quad (9)$$

This decomposition encourages the dynamic core to capture the dominant mass–momentum and thermodynamic evolution, while the Transformer focuses on unresolved nonlinear effects, localized anomalies, and errors induced by the simplified physical approximation.

**Physics-Informed Training Objective.** To encourage physically plausible forecasts, we optimize the model with both data fidelity and weak physical regularization. While the graph-based dynamic core provides an explicit mass–momentum prior through neural ODE integration, the auxiliary losses further constrain the predicted trajectories to respect temporal consistency among meteorological variables.

**1) Data Fidelity.** We supervise predictions using the observed values at each time step:

$$\mathcal{L}_{\text{data}} = \sum_{i,t} \|\hat{X}_{i,t} - X_{i,t}\|_2^2 \quad (10)$$

**2) Pressure–Wind Consistency.** Atmospheric pressure drives wind motion. We enforce temporal alignment between pressure and wind variations:

$$\mathcal{L}_{\text{pw}} = \sum_{i,t} \|\Delta_t P_{i,t} - \alpha_i \Delta_t V_{i,t}\|_2^2 \quad (11)$$

where $P_{i,t}$ and $V_{i,t}$ denote the pressure and wind speed at station $i$ and time $t$, $\Delta_t$ is the first-order temporal difference, and $\alpha_i$ is a station-specific learnable parameter that captures local pressure–wind sensitivity.

**3) Energy-Aware Smoothness.** Meteorological variables evolve smoothly over time; abrupt fluctuations are penalized

using a second-order difference:

$$\mathcal{L}_{\text{smooth}} = \sum_{i,t} \left( \|\Delta_t^2 T_{i,t}\|_2^2 + \|\Delta_t^2 V_{i,t}\|_2^2 \right) \quad (12)$$

where $T_{i,t}$ is the temperature at station $i$ and time $t$, and $\Delta_t^2$ denotes the second-order temporal difference.

The final training objective combines these terms:

$$\mathcal{L} = \mathcal{L}_{\text{data}} + \lambda_1 \mathcal{L}_{\text{pw}} + \lambda_2 \mathcal{L}_{\text{smooth}} \quad (13)$$

where $\lambda_1$ and $\lambda_2$ are the weights of the physical terms.

### 4.4. Evaluation Metrics

**Overall performance.** Mean Absolute Error (MAE) and Mean Squared Error (MSE) are used to evaluate overall GSWF performance. MAE measures the predictive robustness of an algorithm but is insensitive to outliers, whereas MSE is sensitive to outliers and can amplify errors.

**Metric for extreme values.** In addition to standard metrics such as MAE and MSE, the ability to predict extreme weather events, such as unusually high or low temperatures, is crucial in real-world applications. Hence, we introduce a specialized metric, the Symmetric Extremal Dependence Index (SEDI) (Han et al., 2026; Xu et al., 2024). It classifies each prediction at its station location as either extreme or normal weather based on upper quantile thresholds $(90\%, 95\%, 98\%, \text{ and } 99.5\%)$ or lower quantile thresholds $(10\%, 5\%, 2\%, \text{ and } 0.5\%)$. The SEDI value is formulated as:

$$
\begin{aligned}
\text{SEDI}(p) = &\frac{\sum(\hat{X} < Q_L^p \, \& \, X < Q_L^p)}{\sum(X < Q_L^p) + \sum(X > Q_U^p)} \\
&+ \frac{\sum(\hat{X} > Q_U^p \, \& \, X > Q_U^p)}{\sum(X < Q_L^p) + \sum(X > Q_U^p)}
\end{aligned}
\quad (14)
$$

where $\hat{X} < Q_L^p$ and $X < Q_L^p$ indicate whether the predicted or observed data point corresponds to an extreme event under the threshold $Q_L^p$, and vice versa for the upper percentiles. We assess both tails of the distribution to capture extremely low values (*e.g.,* winter storms) and extremely high values (*e.g.,* heatwaves). SEDI $\in [0, 1]$ quantifies the model's ability to correctly identify extreme weather events. Higher SEDI indicates better performance in extreme weather prediction.

### 4.5. Experimental Protocols

**Dataset splitting.** The WEATHER-5K dataset is divided into training (2014–2021), validation (2022), and test (2023) subsets, following an 8:1:1 ratio.

**Task settings.** To enable fair comparison across baselines, we align the input length of all models. Each model predicts the $\tau$-step future based on 48 historical steps, a choice informed by performance trends for four weather variables (Appendix Figure 9) and balancing computation with accuracy. We predict future weather conditions for lead times of 1, 3, 5, and 7 days, corresponding to 24-, 72-, 120-, and 168-step future data. Results are reported once, which suffices due to the dataset's large volume and observed stability across different seeds.

**Implementation details.** Full implementation details, including training settings, hyperparameters, and hardware specifications, are provided in Appendix E.

### 4.6. Observations and Findings

**RQ1: How do TSF models compare with NWP models in terms of general performance?** Overall, as shown in Table 2, the NWP model, *i.e.,* ECMWF-HRES (European Centre for Medium-Range Weather Forecasts, 2024), outperforms all existing TSF methods across nearly all variables, except for wind speed and direction. In Appendix Tables 8 and 9, we further analyze performance across forecast horizons. For short-term prediction (lead time = 24 hours), some TSF methods (**e.g.,** Pyraformer (Liu et al., 2022)) exhibit comparable performance. However, for long-term prediction, cumulative error stabilizes and then increases significantly after the third day. This suggests that predictive errors become substantial beyond three days, indicating considerable room for improvement in long-term forecasting accuracy.

**RQ2: Can TSF models effectively predict extreme weather events as an operational metric?** Table 3 presents the predictive performance of various models on extreme values. Our analysis reveals that TSF models consistently struggle to capture selected extreme thresholds (90% and 99.5%), while numerical models excel at forecasting these extremes, a crucial capability for operational assessments of extreme weather events. Moreover, wind prediction shows the poorest performance among all variables, likely due to the non-stationary nature of wind distributions. These findings underscore the need for future research to enhance TSF models' ability to capture extreme values for more robust operational forecasting.

**RQ3: Are large-parameter TSF models necessary for weather forecasting?** Figure 4 and Table 7 compare the model complexity and prediction accuracy (averaged over 72 hours) of various baselines. Our findings indicate that increasing parameter counts does not necessarily lead to better accuracy. For instance, Pyraformer (Liu et al., 2022) outperforms other models while using a relatively low number of parameters (7.54 MB) and lower training costs. In contrast, Timer (Liu et al., 2024c) and Corrformer (Wu et al.,

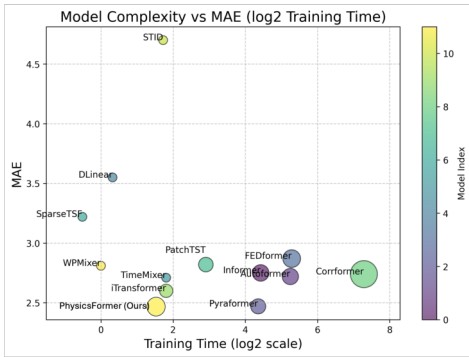

*Figure 4.* Model performance vs complexity. The bubble area represents the parameters in log2 scale.

2023), despite having much higher parameter counts (84 MB and 666 MB), offer little improvement or even worse performance. Although larger models like LTMs may excel in extreme event prediction (see Table 3), Chronos, with only half as many parameters (48 MB), outperforms Timer. Therefore, for practical deployment and frequent updates, smaller models may be more suitable for time series forecasting in weather applications.

**RQ4: How does PhysicsFormer compare with existing TSF and NWP models?** As shown in Table 2, Physics-Former outperforms all baseline TSF methods across most variables and matches or exceeds ECMWF-HRES (European Centre for Medium-Range Weather Forecasts, 2024) in short- and medium-term forecasts. This improvement stems from the dynamic core, which provides physically consistent baseline predictions, and the physics-informed residual learning, which enables the Transformer to correct unresolved patterns while respecting meteorological laws. As Figure 4 shows, our approach achieves strong performance with relatively fewer parameters and training time, demonstrating that combining a physically grounded core with a Transformer backbone and physics-informed loss yields accurate and physically consistent forecasts efficiently.

**RQ5: How can future weather forecasting methods leverage the advantages of TSF and NWP models?** Most NWP models (Bi et al., 2023; Lam et al., 2023; Han et al., 2024) can provide robust global atmospheric forecasts. By using outputs from these models, we can develop bias-correction models tailored to meteorological stations. Bridging GSWF with numerical prediction models could therefore enhance station-level weather prediction accuracy.

### 4.7. Case Study: Heatwave Forecasting

Here we evaluate the performance of several models in forecasting the 2022 summer heatwave in Chongqing. As shown in Figure 5, current time-series forecasting (TSF) methods, such as iTransformer and SparseTSF, still lag behind PhysicsFormer in capturing extreme weather events. PhysicsFormer achieves the lowest MAE (1.23), demonstrat-

ing superior accuracy. However, the Chronos model (MAE 1.82), leveraging a large language model, shows significant potential by narrowing the gap with PhysicsFormer. These results highlight the need for further advancements in TSF methods to improve their effectiveness in extreme event predictions.

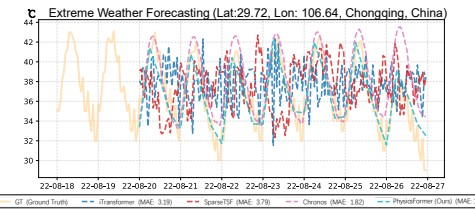

*Figure 5.* Comparison of extreme weather forecasting showcases using several models.

## 5. Conclusion

To support accurate, efficient, and scalable weather forecasting for global stations, we introduce WEATHER-5K as a new benchmark dataset. WEATHER-5K comprises observations from numerous global stations, providing comprehensive, long-term meteorological data. This dataset enables state-of-the-art time-series forecasting methods to be easily adopted and evaluated. However, we observe that current methods still lag behind numerical weather prediction models, particularly for longer lead times. We further propose a physics-informed model that integrates a dynamic core with physics-constraint losses to improve prediction accuracy. WEATHER-5K presents new challenges and opportunities, fostering innovative research.

## Acknowledgements

This research was supported by funding from the Hong Kong RGC General Research Fund (152228/23E, 162161/24E, 162116/25E, and 162180/25E), the National Natural Science Foundation of China (NSFC) Key Program (No. 62532005), the Collaborative Research Fund (Nos. C1042-23GF and C5097-25G), the NSFC/RGC Collaborative Research Scheme (Grant Nos. 62461160332 and CRS_HKUST602/24), the Research Impact Fund (No. R5011-23F), the Areas of Excellence Scheme (AoE/E-601/22-R), and InnoHK (HKGAI).

## Impact Statement

This work aims to advance the development of machine learning models for time-series forecasting in weather prediction. The methods and dataset are intended to improve scientific understanding and forecasting accuracy. The dataset and models may benefit researchers and operational forecasting systems, contributing positively to weather prediction and disaster preparedness.

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

## A. Limitations

**Spatial Quality.** The current version of WEATHER-5K offers observations from a global network, but coverage remains sparse in regions like Africa and South America. Future enhancements will focus on integrating data from diverse sources, such as MADIS reports (including METAR and Mesonet), to achieve denser station distribution and improve evaluation accuracy.

**Missing Data Handling.** To maintain data integrity, we used ERA5 interpolation to fill missing observations. Although this interpolation introduces some errors, with a limited impact on temperature, dew point temperature, and pressure but a larger impact on wind data, only a small fraction of missing values were filled to minimize these effects.

## B. Observations and Future Directions

After a comprehensive evaluation, our benchmark reveals that current TSF methods lag behind operational NWP models, particularly in long-term forecasting and extreme event prediction as shown in Figure 1. To bridge these gaps, we identify several critical future directions for advancing TSF research for operational weather forecasting, based on insights from our dataset and benchmarks:

- Improve overall weather forecasting performance by prioritizing *long-term forecast* accuracy.

- Address the significant limitations in predicting *extreme weather events* by focusing research on this critical area.

- Our benchmarks indicate that **larger** models with more parameters *do not* consistently yield better performance; developing lightweight, efficient TSF models may be more beneficial.

- Some metrics (*e.g.,* wind rate) suggest that TSF can outperform NWP. A *hybrid forecasting model* that enjoys the simplicity of TSF with the operational strengths of NWP may offer a promising solution.

In summary, our work provides valuable resources, benchmarks, and guidance to propel TSF research in weather forecasting. We believe it will enable researchers to better evaluate and compare algorithms, fostering the development of more accurate and reliable forecasting techniques for global weather systems.

## C. Dataset Documentation

We organize the dataset documentation based on the template of datasheets for datasets (Gebru et al., 2021).

### C.1. Motivation

**For what purpose was the dataset created? Was there a specific task in mind? Was there a specific gap that needed to be filled? Please provide a description.**

This dataset was created with the following motivations. 1) Existing weather station datasets limit the applicability of forecasting models in real-world scenarios. Therefore, a comprehensive global station weather dataset is needed to enable forecasting models to generalize across diverse stations and regions worldwide. 2) The limited size of existing time-series datasets may not reflect the real performance of forecasting models; the proposed large-scale weather station dataset can also serve as an extensive time-series benchmark for various forecasting methods. 3) Existing simple datasets fail to encompass the complex scientific problems that researchers need to discover and resolve, thereby hindering progress in time-series prediction. The proposed dataset offers diverse temporal and spatial coverage, enabling comprehensive evaluation of time-series forecasting methods and driving progress in the field. 4) A large-scale weather station dataset is a crucial source of observational data for numerical weather prediction models, bridging the gap between numerical models and station-based predictions. This not only improves the accuracy of numerical forecasts but also plays a vital role in verifying and evaluating numerical weather prediction models.

### C.2. Composition

**What do the instances that comprise the dataset represent (e.g., documents, photos, people, countries)? Are there multiple types of instances (e.g., movies, users, and ratings; people and interactions between them; nodes and edges)?**

**Please provide a description.**

WEATHER-5K consists of 5,672 CSV files. Each CSV file represents data from a single weather station, with hourly observations recorded from 2014 to 2023. The dataset represents a collection of weather observation data, where each instance corresponds to an hourly observation from a specific weather station, with various meteorological measurements and auxiliary information.

**How many instances are there in total (of each type, if appropriate)?**

WEATHER-5K contains 5,672 stations with 10 years of temporal coverage. Each station has 87,648 hourly records and includes mandatory meteorological variables and auxiliary fields.

**Does the dataset contain all possible instances or is it a sample (not necessarily random) of instances from a larger set? If the dataset is a sample, then what is the larger set? Is the sample representative of the larger set (e.g., geographic coverage)? If so, please describe how this representativeness was validated/verified.**

Our dataset is collected and further processed using data sourced from the National Centers for Environmental Information (NCEI), specifically the Integrated Surface Database (ISD), [2]. Although the ISD contains records from over $20,000$ stations spanning several decades, not all stations are suitable for machine learning applications. For instance, some stations are no longer operational, many do not report data on an hourly basis, and numerous stations have missing values for critical weather elements. To obtain a high-quality weather station dataset, a meticulous selection process was conducted to include only long-term, hourly reporting stations that are currently operational and provide essential observations such as temperature, dew point temperature, wind, and sea level pressure. We then used the processing procedure detailed in Section 3 to construct the final WEATHER-5K dataset, ensuring its applicability to time-series forecasting research.

**What data does each instance consist of? "Raw" data (e.g., unprocessed text or images) or features? In either case, please provide a description.**

The key characteristics of each instance are:

1. **Instance Type**: Each row in the CSV file represents a single hourly weather observation from a specific weather station.

2. **Instance Fields:** Each instance (row) contains the fields as detailed in Table 4.

3. **Temporal Dimension:** The dataset covers hourly weather observations from 2014-01-01 T00:00:00 to 2023-12-31 T00:00:00, a total of 87,648 time slots.

4. **Spatial Dimension:** Each CSV file represents data from a single weather station, identified by its geographic coordinates (`LONGITUDE` and `LATITUDE`).

*Table 4.* Instance Fields.

| Field | Description |
| --- | --- |
| DATE | The date of the observation |
| LONGITUDE | The longitude of the weather station |
| LATITUDE | The latitude of the weather station |
| TMP | The temperature observation |
| DEW | The dew point observation |
| WND_ANGLE | The wind angle observation |
| WND_RATE | The wind rate observation |
| SLP | The sea level pressure observation |
| MASK | A binary list indicating the quality of the observation |
| TIME_DIFF | An auxiliary field |

**Is there a label or target associated with each instance? If so, please provide a description.**

No. Weather observation data serve as their own targets in the forecasting task, and weather forecasting can be considered a self-supervised learning task.

---

[2] www.ncei.noaa.gov/products/land-based-station/integrated-surface-database

**Is any information missing from individual instances? If so, please provide a description, explaining why this information is missing (e.g., because it was unavailable).**

No. Substantial effort has been made to ensure that there are no missing values in the WEATHER-5K dataset.

**Are relationships between individual instances made explicit (e.g., users' movie ratings, social network links)? If so, please describe how these relationships are made explicit.**

Yes, the weather stations in the dataset have geographical relationships, and we have used latitude, longitude, and elevation to represent their geographic locations. This information can be leveraged in subsequent work to model the spatial relationships between the instances.

**Are there recommended data splits (e.g., training, development/validation, testing)? If so, please provide a description of these splits, explaining the rationale behind them.**

Yes, we chronologically split the data into training (2014-01-01 to 2021-12-31), validation (2022-01-01 to 2022-12-31), and test (2023-01-01 to 2023-12-31) sets, with a ratio of 8:1:1.

**Are there any errors, sources of noise, or redundancies in the dataset? If so, please provide a description.**

Yes, the errors and noise in the dataset arise from two main sources. Firstly, the use of meteorological automatic stations introduces a certain degree of observational error, particularly in the measurement of wind speed and direction, which are relatively difficult to measure accurately. Secondly, in our data processing efforts to ensure the completeness of the dataset, we have employed interpolation operations, which can introduce some additional error. However, the proportion of error introduced by the interpolation is relatively small.

**Is the dataset self-contained, or does it link to or otherwise rely on external resources (e.g., websites, tweets, other datasets)?**

Yes, it is self-contained.

**Does the dataset contain data that might be considered confidential (e.g., data that is protected by legal privilege or by doctor–patient confidentiality, data that includes the content of individuals' non-public communications)? If so, please provide a description.**

No, all our data are from a publicly available data source, i.e., NCEI.

**Does the dataset contain data that, if viewed directly, might be offensive, insulting, threatening, or might otherwise cause anxiety? If so, please describe why.**

No, all our data are numerical.

### C.3. Collection Process

**How was the data associated with each instance acquired? Was the data directly observable (e.g., raw text, movie ratings), reported by subjects (e.g., survey responses), or indirectly inferred/derived from other data (e.g., part-of-speech tags, model-based guesses for age or language)?**

We source the data from the Integrated Surface Database (ISD), organized and maintained by the National Centers for Environmental Information (NCEI). ISD is a global database that consists of hourly and synoptic surface observations compiled from numerous sources into a single common ASCII format and common data model. ISD integrates data from more than 100 original data sources.

**What mechanisms or procedures were used to collect the data (e.g., hardware apparatuses or sensors, manual human curation, software programs, software APIs)? How were these mechanisms or procedures validated?**

NCEI (formerly the National Climatic Data Center) started developing ISD in 1998 with assistance from partners in the U.S. Air Force and Navy, as well as external funding from several sources. The database incorporates data from over 35,000 stations around the world, with some observation records dating back to 1901. The number of stations with data in ISD increased substantially in the 1940s and again in the early 1970s. There are currently more than 14,000 active ISD stations that are updated daily in the database. The total uncompressed data volume is around 600 gigabytes; however, it continues to grow as more data are added.

**If the dataset is a sample from a larger set, what was the sampling strategy (e.g., deterministic, probabilistic with specific sampling probabilities)?**

The sampling strategy is deterministic.

**Who was involved in the data collection process (e.g., students, crowdworkers, contractors) and how were they compensated (e.g., how much were crowdworkers paid)?**

Our code collects publicly available data, which is free. On our side, we developed a download API to efficiently retrieve the source data, which was done by our team members.

**Over what timeframe was the data collected? Does this timeframe match the creation timeframe of the data associated with the instances (e.g., recent crawl of old news articles)? If not, please describe the timeframe in which the data associated with the instances was created.**

The WEATHER-5K dataset was collected and processed in 2024. The timeframe of the source data matches the creation timeframe of the data.

**Were any ethical review processes conducted (e.g., by an institutional review board)?**

No, such processes are unnecessary in our case.

### C.4. Preprocessing/cleaning/labeling

**Was any preprocessing/cleaning/labeling of the data done (e.g., discretization or bucketing, tokenization, part-of-speech tagging, SIFT feature extraction, removal of instances, processing of missing values)? If so, please provide a description.**

Yes. To obtain a high-quality dataset of weather stations, a series of post-processing steps were performed on the raw weather station data collected from 2014 to 2024. Initially, $10,701$ commonly operating stations were identified. The first step involved selecting stations that reported data every hour on the hour. However, many stations did not meet this criterion. To address this, a replacement method estimated missing hourly data points using the nearest available time points within a 30-minute window, significantly improving the distribution of valid hourly data. Additional processing steps are described in Section 3.

**Was the "raw" data saved in addition to the preprocessed / cleaned/ labeled data (e.g., to support unanticipated future uses)? If so, please provide a link or other access point to the "raw" data.**

The raw data are available from NCEI. The link is: www.ncei.noaa.gov/products/land-based-station/integrated-surface-database. To obtain the preprocessed data, users can run weather_station_api.py in our released repository.

**Is the software that was used to preprocess/clean/label the data available? If so, please provide a link or other access point.**

No.

### C.5. Uses

**Has the dataset been used for any tasks already? If so, please provide a description.**

The dataset is used in this paper for the global station weather forecasting task.

**Is there a repository that links to any or all papers or systems that use the dataset? If so, please provide a link or other access point.**

No, but we may include a leaderboard and list papers using this dataset in the future.

**What (other) tasks could the dataset be used for?**

Weather data imputation, numerical weather prediction, and data assimilation.

**Is there anything about the composition of the dataset or the way it was collected and preprocessed/cleaned/labeled that might impact future uses?**

We believe that our dataset does not impose usage limitations.

**Are there tasks for which the dataset should not be used? If so, please provide a description.**

No. Users may use our dataset for any task as long as it does not violate applicable laws.

### C.6. Distribution

**Will the dataset be distributed to third parties outside of the entity (e.g., company, institution, organization) on behalf of which the dataset was created? If so, please provide a description.**

No, it will always be held on GitHub.

**How will the dataset be distributed (e.g., tarball on website, API, GitHub)? Does the dataset have a digital object identifier (DOI)?**

The instructions for building WEATHER-5K will be available in the released code. The dataset does not have a digital object identifier currently.

**When will the dataset be distributed?**

On June 7, 2024.

**Will the dataset be distributed under a copyright or other intellectual property (IP) license, and/or under applicable terms of use (ToU)? If so, please describe this license and/or ToU, and provide a link or other access point to.**

Our benchmark dataset is released under a CC BY-NC 4.0 International License: https://creativecommons.org/licenses/by-nc/4.0. Our code implementation is released under the MIT License: https://opensource.org/licenses/MIT.

**Have any third parties imposed IP-based or other restrictions on the data associated with the instances? If so, please describe these restrictions, and provide a link or other access point to, or otherwise reproduce, any relevant licensing terms, as well as any fees associated with these restrictions.**

Yes, for commercial use, please check the website: https://www.ncei.noaa.gov/.

**Do any export controls or other regulatory restrictions apply to the dataset or to individual instances? If so, please describe these restrictions, and provide a link or other access point to, or otherwise reproduce, any supporting documentation.** No.

### C.7. Maintenance

**Who will be supporting/hosting/maintaining the dataset?**

The authors of the paper.

**Is there an erratum? If so, please provide a link or other access point.**

Users can use GitHub to report issues or bugs.

**Will the dataset be updated (e.g., to correct labeling errors, add new instances, delete instances)? If so, please describe how often, by whom, and how updates will be communicated to dataset consumers (e.g., mailing list, GitHub)?**

Yes, the authors will actively update the code and data on GitHub. Any updates to the dataset will be announced in our GitHub repository.

**If the dataset relates to people, are there applicable limits on the retention of the data associated with the instances (e.g., were the individuals in question told that their data would be retained for a fixed period of time and then deleted)? If so, please describe these limits and explain how they will be enforced.**

The dataset does not relate to people.

**Will older versions of the dataset continue to be supported/ hosted/ maintained? If so, please describe how. If not, please describe how its obsolescence will be communicated to dataset consumers.**

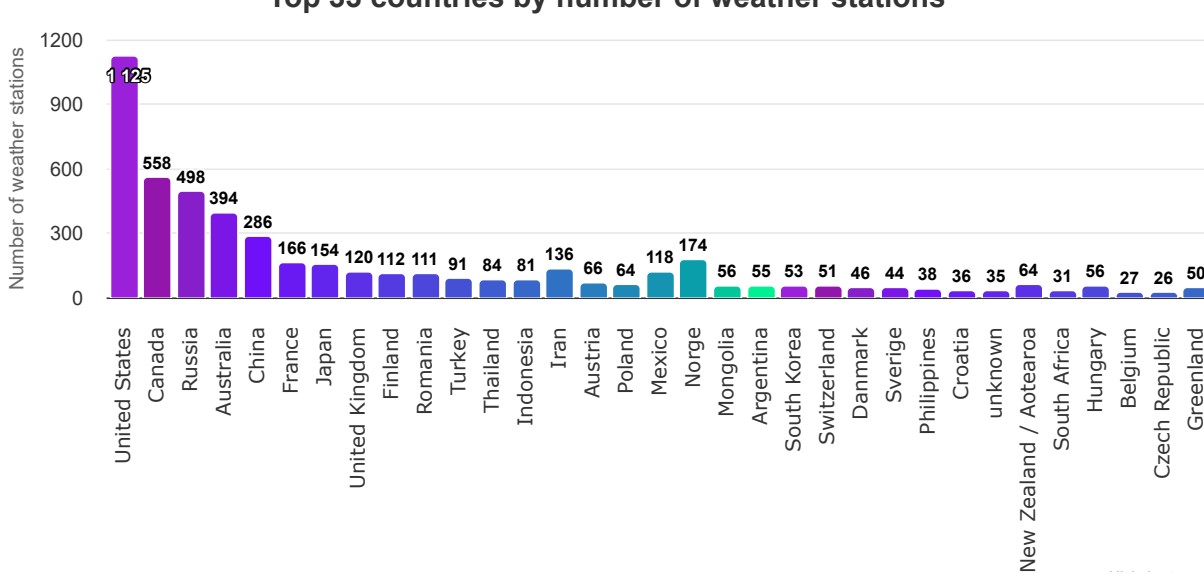

*Figure 6.* Statistics on the number of weather stations in different countries and regions.

Yes, we will provide the information on GitHub.

**If others want to extend/augment/build on/contribute to the dataset, is there a mechanism for them to do so? If so, please provide a description. Will these contributions be validated/verified? If so, please describe how. If not, why not? Is there a process for communicating/distributing these contributions to dataset consumers? If so, please provide a description.**

Yes, we welcome users to submit pull requests on GitHub, and we will actively validate the requests.

## D. More Dataset Analysis

**Disparities of station density.** The WEATHER-5K dataset reveals regional disparities in the distribution of weather stations, which can significantly impact the learning and understanding of atmospheric dynamics in certain areas. As illustrated in Figure 7 **a)**, some land regions have sparse data coverage compared to others. These disparities can be attributed to factors such as geographical characteristics, levels of economic development, and the strategic placement of weather stations. Note that the number of oceanic stations is also very limited due to the expensive cost of establishing stations at sea. Addressing these disparities and expanding coverage in underrepresented areas is crucial for improving the accuracy and reliability of weather forecasting and analysis in those regions.

**Different data patterns.** By visualizing the temperature observations, as shown in Figure 7 **b)**, *temperature* shows a seasonal pattern. However, wind speeds are non-stationary, characterized by intense fluctuations and a lack of clear patterns, making them challenging to predict.

**Comparison between WEATHER-5K and ERA5.** ERA5 is a simulated dataset, not based on in-situ observations, which limits its applicability in real-world scenarios. In contrast, the WEATHER-5K dataset is derived from in-situ observations. In addition to the point-based comparison, Figure 7 **c)** and Figure 7 **d)** also demonstrate the relationship and differences between reanalysis data and observations.

**Station distribution by country.** Figure 6 shows the histogram of the number of weather stations in the WEATHER-5K dataset over 33 countries. WEATHER-5K is a global database, though the best spatial coverage is evident in North America, Europe, Australia, and parts of Asia. Coverage in the Northern Hemisphere is better than the Southern Hemisphere.

**Comparison between ISD and MADIS data sources.** Overall, as shown in Table 5, ISD is more diverse and offers broader

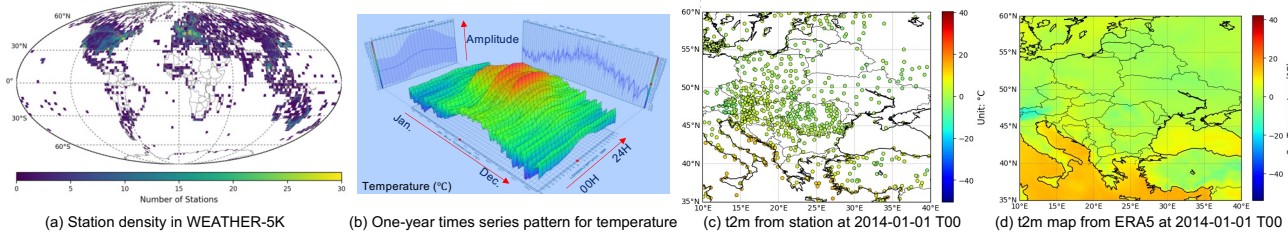

(a) Station density in WEATHER-5K    (b) One-year times series pattern for temperature    (c) t2m from station at 2014-01-01 T00    (d) t2m map from ERA5 at 2014-01-01 T00

*Figure 7.* Additional data analysis and visualizations.

coverage. Specifically, the surface data in MADIS mainly include METAR [3] and Mesonet [4]. Its reports primarily come from the U.S., whereas ISD collects surface weather data from more than 35, 000 stations worldwide. Additionally, ISD spans a longer period, from 1901 to the present, and is fully public for users. In future research, we believe including MADIS for station-based weather forecasting will further enhance this field.

*Table 5.* Differences between ISD (database of WEATHER-5K) and MADIS

| Dataset | Availability | Data Source | Time | Coverage |
|---|---|---|---|---|
| MADIS (METAR and Mesonet) | Restricted | ASOS, AWOS, Airport Reports, CWOP, FAWH, GPSMet, KSDOT, RAWS, UDFCD, GLDNWS, IADOT, INTERNET | 2001-present | Primarily in U.S. |
| ISD | Fully public | More than 100 original data sources | 1901-present | 35,000 global stations |

*Table 6.* Statistics of WEATHER-5K dataset.

|  | Temperature | Dewpoint | Wind Direction | Wind Speed | Sea Level Pressure |
|---|---|---|---|---|---|
| Mean | 12.71 | 6.53 | 191.19 | 3.37 | 1014.85 |
| Standard Deviation | 13.08 | 12.14 | 99.67 | 2.66 | 9.17 |

**Characteristics of data distribution**. Figure 8 provides violin plots for several variables. For temperature and dewpoint, the distributions of their data have similar shapes. The upper and lower distributions of the data are symmetrical around the median. The temperature distribution is most concentrated in the low-latitude regions. As the latitude increases, the center of the temperature distribution starts to shift and also becomes more dispersed. This indicates that the temperature difference is larger in the mid-to-high latitude regions. For sea level pressure, we find that the distribution centers are similar across different latitudes, with little shift. However, as the latitude increases, the dispersion of sea level pressure becomes greater.

**Climate mean and standard deviation**. Table 6 presents the mean and standard deviation values for five key weather variables measured at 5,672 weather stations. The variables included are temperature, dewpoint, wind direction, wind speed, and sea level pressure. The mean temperature across the weather stations is 12.71 degrees, with a standard deviation of 13.08 degrees. For dewpoint, the mean is 6.53 degrees and the standard deviation is 12.14 degrees. The mean wind direction is 191.19 degrees, with a standard deviation of 99.67 degrees. The mean wind speed is 3.37 meters per second, with a standard deviation of 2.66 meters per second. Finally, the mean sea level pressure is 1014.85 millibars, with a standard deviation of 9.17 millibars. These statistics provide a high-level overview of the typical weather conditions captured by the network of weather stations.

---

[3] https://madis.ncep.noaa.gov/madis_metar.shtml
[4] https://madis.ncep.noaa.gov/madis_mesonet.shtml

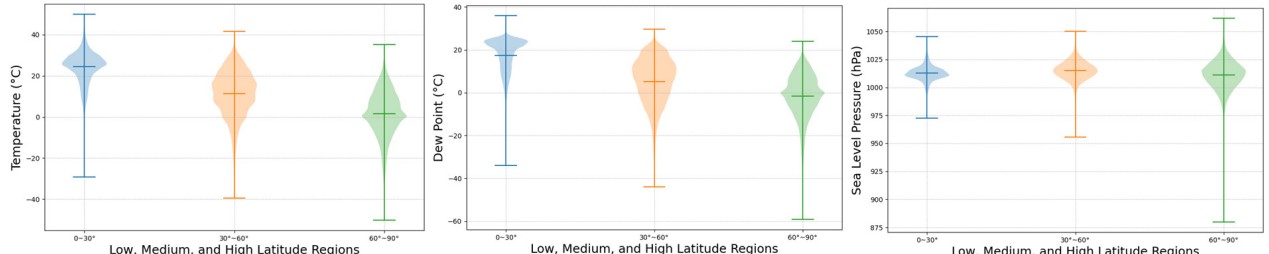

*Figure 8.* Violin plots of temperature, dew point, and sea level pressure observations in low, middle, and high latitudes.

*Table 7.* Efficiency comparisons. Statistics are reported under the following setting: batch size → 1024 in general, except 5600 for Corrformer, MTGNN, and STID; train time → estimated training time (in hours) for 300,000 iterations; task setting → 48 input steps and 120 output steps; hardware → a computing server equipped with 224 Intel(R) Xeon(R) Platinum 8480CL CPUs @ 3.80 GHz, 2.0 TB RAM, and 8 NVIDIA H800 GPUs. Each experiment is conducted on a single GPU. The total training time is also influenced by data-read speed, as some methods may suffer from input starvation.

|  | Informer | Autoformer | Pyraformer | FEDformer | DLinear | TimeMixer |
|---|---|---|---|---|---|---|
| Training Time (Hours) | 21∼22 | 36∼40 | 20 ∼ 21 | 38 ∼ 40 | 1.0 ∼ 1.5 | 3.5 |
| GPU Memory (MiB) | 12,880 | 64,688 | 33,750 | 18,804 | 850 | 1,191 |
| Parameters (M) | 11.32 | 10.53 | 7.54 | 16.29 | 0.01 | 0.06 |

|  | PatchTST | Corrformer | iTransformer | MTGNN | STID | WPMixer |
|---|---|---|---|---|---|---|
| Training Time (Hours) | 7∼8 | 144∼168 | 3∼4 | 35 | 3.3 | 1 |
| GPU Memory (MiB) | 22,512 | 46,486 | 45,672 | 5,155 | 813 | 1,499 |
| Parameters (M) | 6.65 | 666.12 | 4.8 | 40.06 | 0.24 | 0.05 |

|  | SparseTSF | Timer | Chronos | PhysicsFormer (Ours) |
|---|---|---|---|---|
| Training Time (Hours) | 0.7 | / | / | 3 |
| GPU Memory (MiB) | 525 | / | / | 9318 |
| Parameters (M) | 0.0002 | 84 | 48 | 19.33 |

# E. Implementation Details and Additional Experimental Results

**Implementation details.** We develop and implement the baselines based on the Time-Series-Library (THUML, 2024). Training is performed for 300,000 iterations, starting with a learning rate of 1e-4. We employ a cosine decay strategy and gradually decay the learning rate to 0 by the end of training. The batch size for all models is set to 1,024, except for Corrformer, STID, and MTGNN. During validation, early stopping is triggered if the training loss does not decrease for three consecutive checks. The checkpoint with the lowest validation loss before early stopping is saved and used for testing. Experiments are conducted on a computing server equipped with 224 Intel(R) Xeon(R) Platinum 8480CL CPUs @ 3.80 GHz, 2.0 TB RAM, and 8 NVIDIA H800 GPUs.

**Full benchmark results.** We present the full forecasting metrics for all methods at four different prediction lengths in Table 8 and Table 9. We also include Mamba-based methods (WSSM (Yang et al., 2025a)) and time-series generation methods (UniMTD (Yang et al., 2025b), UniTraj (Xu & Fu, 2025)) in the table.

**Ablation study.** The ablation experiments explore the impact of different input lengths on predictions. We use the iTransformer (Liu et al., 2024a) model for verification. Specifically, we set the input length to $24, 48, 72, 96$, and $120$, while keeping the prediction length fixed at 72. The experimental results are shown in Figure 9, illustrating the variation in MAE for four weather variables as the input sequence length increases. The results demonstrate that performance for some variables (temperature and wind) improves slightly when the input length increases. In contrast, the MAE for sea level pressure rises slightly. We ultimately set the input length to 48 to balance computation and performance.

**Efficiency comparisons.** We summarize the efficiency comparisons in Table 7, which details the complexity metrics, with LTMs employing zero-shot prediction.

**Visualization results.** In Figures 10, 11, 12, 13, 14, 15, 16, 17, 18, 19, 20, and 21, we plot visualization results to showcase the performance of various time-series forecasting methods, including Pyraformer, FEDformer, DLinear, PatchTST,

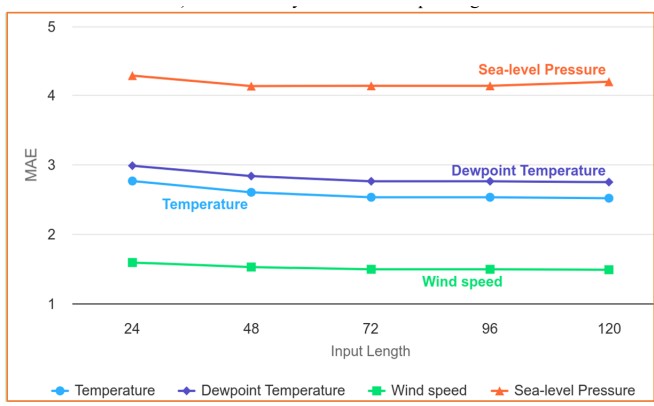

*Figure 9.* Ablation study on input length.

iTransformer, Corrformer, STID, SparseTSF, TimeMixer, Chronos, Timer, and PhysicsFormer. These visualizations provide a comparative analysis of how different forecasting approaches perform on the time-series data.

This series of figures illustrates the characteristics and capabilities of each method, helping readers understand the strengths and weaknesses of the various techniques and how they may be suited for different types of time-series forecasting problems.

*Table 8.* Full results of all the baseline models. (Part I)

| Methods | Lead Time | Temperature MAE | Temperature MSE | Dewpoint MAE | Dewpoint MSE | Wind Rate MAE | Wind Rate MSE | Wind Direc. MAE | Wind Direc. MSE | Sea Level Pressure MAE | Sea Level Pressure MSE |
|---|---|---|---|---|---|---|---|---|---|---|---|
| ECMWF-HRES | 24 | 1.76 | 7.39 | 1.85 | 7.94 | 1.48 | 4.53 | 63.8 | 7158.3 | 0.86 | 2.68 |
| | 72 | 1.87 | 8.01 | 1.94 | 8.48 | 1.52 | 4.76 | 72.4 | 8215.6 | 1.06 | 3.31 |
| | 120 | 1.99 | 8.79 | 2.14 | 10.87 | 1.58 | 5.11 | 75.4 | 8647.7 | 1.38 | 5.15 |
| | 168 | 2.15 | 10.06 | 2.32 | 12.56 | 1.66 | 5.59 | 78.3 | 8945.7 | 1.87 | 9.52 |
| Pyraformer | 24 | 1.75 | 6.92 | 1.83 | 7.88 | 1.30 | 3.58 | 61.8 | 6930.2 | 1.90 | 9.72 |
| | 72 | 2.47 | 13.03 | 2.67 | 15.39 | 1.52 | 4.97 | 72.0 | 8222.4 | 3.76 | 33.67 |
| | 120 | 2.77 | 16.04 | 3.00 | 18.95 | 1.59 | 5.37 | 75.1 | 8610.7 | 4.43 | 43.91 |
| | 168 | 2.95 | 17.95 | 3.20 | 21.06 | 1.61 | 5.56 | 76.4 | 8773.5 | 4.77 | 49.97 |
| Informer | 24 | 1.88 | 7.51 | 1.94 | 8.30 | 1.30 | 3.62 | 60.7 | 6906.9 | 2.01 | 10.56 |
| | 72 | 2.75 | 14.84 | 2.86 | 17.24 | 1.53 | 4.86 | 71.5 | 8251.4 | 4.24 | 39.24 |
| | 120 | 3.11 | 18.21 | 3.25 | 21.50 | 1.60 | 5.38 | 75.7 | 8504.5 | 5.15 | 54.31 |
| | 168 | 3.24 | 20.24 | 3.43 | 24.89 | 1.63 | 5.65 | 76.2 | 8718.4 | 5.26 | 58.42 |
| Autoformer | 24 | 1.93 | 8.64 | 2.06 | 9.57 | 1.42 | 3.97 | 66.5 | 7710.0 | 2.26 | 12.78 |
| | 72 | 2.72 | 15.14 | 2.97 | 18.38 | 1.54 | 5.14 | 75.4 | 9111.5 | 4.25 | 42.34 |
| | 120 | 3.21 | 20.27 | 3.34 | 23.12 | 1.58 | 5.73 | 79.2 | 9143.5 | 4.83 | 48.88 |
| | 168 | 3.43 | 21.71 | 3.56 | 22.55 | 1.64 | 5.95 | 79.8 | 9435.8 | 5.32 | 61.85 |
| FEDformer | 24 | 1.98 | 8.45 | 2.02 | 9.25 | 1.36 | 3.91 | 66.0 | 7384.1 | 2.13 | 11.43 |
| | 72 | 2.87 | 16.50 | 3.01 | 18.70 | 1.59 | 5.31 | 76.2 | 8824.8 | 4.15 | 37.60 |
| | 120 | 3.19 | 20.29 | 3.36 | 23.10 | 1.66 | 5.71 | 79.0 | 9143.3 | 4.81 | 48.86 |
| | 168 | 3.35 | 22.12 | 3.54 | 25.21 | 1.68 | 5.88 | 79.7 | 9189.2 | 5.01 | 53.39 |
| PatchTST | 24 | 2.05 | 9.26 | 2.16 | 10.58 | 1.40 | 4.20 | 66.2 | 7765.8 | 2.19 | 12.54 |
| | 72 | 2.82 | 16.60 | 3.06 | 19.96 | 1.60 | 5.39 | 75.2 | 9067.8 | 4.28 | 42.46 |
| | 120 | 3.15 | 20.32 | 3.43 | 24.39 | 1.66 | 5.79 | 77.8 | 9452.6 | 5.09 | 57.29 |
| | 168 | 3.33 | 22.54 | 3.63 | 26.94 | 1.69 | 6.00 | 79.0 | 9638.1 | 5.51 | 65.30 |
| Corrformer | 24 | 1.99 | 8.21 | 2.09 | 9.47 | 1.38 | 3.83 | 66.7 | 7832.3 | 2.19 | 12.39 |
| | 72 | 2.74 | 15.16 | 2.99 | 18.40 | 1.56 | 4.91 | 75.6 | 9111.7 | 4.27 | 42.36 |
| | 120 | 3.06 | 18.63 | 3.34 | 22.48 | 1.61 | 5.56 | 78.0 | 9477.4 | 5.08 | 57.13 |
| | 168 | 3.09 | 18.69 | 3.36 | 22.53 | 1.63 | 5.69 | 78.9 | 9636.0 | 5.34 | 61.83 |
| iTransformer | 24 | 1.82 | 7.49 | 1.93 | 8.80 | 1.32 | 3.77 | 63.2 | 7358.8 | 1.99 | 10.84 |
| | 72 | 2.60 | 14.46 | 2.84 | 17.50 | 1.52 | 4.96 | 73.2 | 8713.3 | 4.14 | 40.65 |
| | 120 | 2.97 | 18.36 | 3.24 | 22.16 | 1.59 | 5.42 | 76.4 | 9192.2 | 4.95 | 54.67 |
| | 168 | 3.18 | 20.64 | 3.48 | 24.89 | 1.64 | 5.67 | 78.0 | 9441.1 | 5.36 | 62.31 |
| STID | 24 | 4.65 | 40.31 | 4.17 | 34.36 | 1.69 | 6.27 | 79.2 | 9573.3 | 5.69 | 62.96 |
| | 72 | 4.70 | 41.18 | 4.20 | 34.97 | 1.69 | 6.28 | 79.4 | 9578.2 | 5.77 | 64.47 |
| | 120 | 4.74 | 41.73 | 4.24 | 35.24 | 1.69 | 6.30 | 79.5 | 9556.8 | 5.80 | 65.22 |
| | 168 | 4.77 | 42.39 | 4.26 | 35.67 | 1.69 | 6.30 | 79.5 | 9556.4 | 5.83 | 65.89 |
| Dlinear | 24 | 2.71 | 13.82 | 2.47 | 12.36 | 1.44 | 4.34 | 66.6 | 8234.5 | 3.09 | 21.34 |
| | 72 | 3.55 | 23.05 | 3.48 | 22.85 | 1.62 | 5.37 | 75.0 | 9250.8 | 4.64 | 45.83 |
| | 120 | 3.90 | 27.60 | 3.89 | 27.72 | 1.67 | 5.70 | 77.3 | 9510.6 | 5.19 | 56.22 |
| | 168 | 4.11 | 30.38 | 4.11 | 30.58 | 1.69 | 5.88 | 78.4 | 9630.0 | 5.48 | 61.73 |
| SparseTSF | 24 | 2.63 | 13.16 | 2.32 | 11.96 | 1.49 | 4.67 | 68.4 | 8655.3 | 3.36 | 24.82 |
| | 72 | 3.22 | 19.93 | 3.13 | 20.88 | 1.66 | 5.82 | 76.5 | 9905.2 | 5.00 | 53.40 |
| | 120 | 3.48 | 23.42 | 3.47 | 25.16 | 1.72 | 6.24 | 78.9 | 10290.3 | 5.63 | 66.66 |
| | 168 | 3.64 | 25.63 | 3.66 | 27.71 | 1.75 | 6.47 | 80.1 | 10485.0 | 5.96 | 73.92 |
| TimeMixer | 24 | 2.11 | 9.80 | 2.08 | 10.39 | 1.43 | 4.40 | 67.3 | 8218.1 | 2.28 | 13.37 |
| | 72 | 2.71 | 15.38 | 2.88 | 18.01 | 1.54 | 5.08 | 74.3 | 8758.5 | 4.14 | 40.05 |
| | 120 | 2.92 | 17.58 | 3.14 | 20.78 | 1.56 | 5.22 | 75.8 | 8809.3 | 4.78 | 50.49 |
| | 168 | 3.03 | 18.76 | 3.27 | 22.08 | 1.57 | 5.25 | 76.4 | 8814.9 | 5.04 | 54.87 |
| WPMixer | 24 | 2.21 | 10.65 | 2.19 | 11.15 | 1.49 | 4.75 | 69.7 | 8587.9 | 2.56 | 15.65 |
| | 72 | 2.81 | 16.51 | 2.97 | 19.02 | 1.58 | 5.31 | 75.4 | 9092.2 | 4.26 | 41.90 |
| | 120 | 3.02 | 18.71 | 3.24 | 21.82 | 1.59 | 5.36 | 76.8 | 8906.8 | 4.85 | 51.86 |
| | 168 | 3.14 | 20.12 | 3.38 | 23.40 | 1.59 | 5.38 | 77.1 | 8908.4 | 5.12 | 56.33 |

*Table 9.* Full results of all the baseline models. (Part II)

| Methods | Lead Time | Temperature | | Dewpoint | | Wind Rate | | Wind Direc. | | Sea Level Pressure | |
|---|---|---|---|---|---|---|---|---|---|---|---|
| | | MAE | MSE | MAE | MSE | MAE | MSE | MAE | MSE | MAE | MSE |
| MTGNN | 24 | 10.53 | 179.46 | 9.71 | 155.53 | 2.15 | 8.66 | 88.0 | 10789.2 | 7.20 | 93.48 |
| | 72 | 10.30 | 170.35 | 9.50 | 147.95 | 2.09 | 8.19 | 86.1 | 10136.9 | 6.95 | 87.35 |
| | 120 | 10.26 | 168.87 | 9.47 | 146.95 | 2.08 | 8.13 | 85.9 | 10056.5 | 6.92 | 86.62 |
| | 168 | 10.24 | 168.29 | 9.46 | 146.52 | 2.08 | 8.14 | 85.9 | 10062.6 | 6.92 | 86.59 |
| Chronos | 24 | 2.19 | 11.20 | 2.21 | 11.94 | 1.50 | 5.08 | 68.8 | 9864.8 | 2.34 | 14.86 |
| | 72 | 3.00 | 19.97 | 3.26 | 23.75 | 1.74 | 6.76 | 80.8 | 12045.2 | 4.58 | 50.14 |
| | 120 | 3.36 | 24.52 | 3.70 | 29.65 | 1.80 | 7.23 | 84.7 | 12783.9 | 5.54 | 69.72 |
| | 168 | 3.58 | 27.36 | 3.94 | 32.94 | 1.83 | 7.47 | 86.5 | 13135.8 | 6.01 | 79.62 |
| Timer | 24 | 2.27 | 11.11 | 2.30 | 11.85 | 1.45 | 4.45 | 67.9 | 8287.0 | 2.87 | 19.77 |
| | 72 | 2.96 | 18.25 | 3.14 | 20.92 | 1.63 | 5.61 | 75.7 | 9546.6 | 4.79 | 51.24 |
| | 120 | 3.24 | 21.57 | 3.44 | 24.53 | 1.68 | 5.97 | 77.7 | 9845.7 | 5.45 | 64.30 |
| | 168 | 3.41 | 23.60 | 3.61 | 26.73 | 1.70 | 6.19 | 78.8 | 10013.4 | 5.79 | 71.72 |
| WSSM | 24 | - | - | - | - | - | - | - | - | - | - |
| | 72 | 2.91 | 17.50 | 3.07 | 20.61 | 1.60 | 5.59 | 73.6 | 8660.48 | 4.19 | 40.64 |
| | 120 | 3.17 | 20.68 | 3.46 | 24.81 | 1.68 | 5.98 | 76.0 | 9358.91 | 4.96 | 54.92 |
| | 168 | - | - | - | - | - | - | - | - | - | - |
| UniMTD | 24 | - | - | - | - | - | - | - | - | - | - |
| | 72 | - | - | - | - | - | - | - | - | - | - |
| | 120 | 2.03 | 8.66 | 2.03 | 9.31 | 1.37 | 4.03 | 65.95 | 8059.31 | 2.59 | 16.63 |
| | 168 | - | - | - | - | - | - | - | - | - | - |
| UniTraj | 24 | - | - | - | - | - | - | - | - | - | - |
| | 72 | - | - | - | - | - | - | - | - | - | - |
| | 120 | 2.22 | 8.08 | 2.12 | 8.11 | 1.71 | 4.93 | 84.41 | 10943.16 | 1.57 | 4.66 |
| | 168 | - | - | - | - | - | - | - | - | - | - |
| PhysicsFormer (Ours) | 24 | 1.55 | 5.36 | 1.59 | 6.02 | 1.21 | 3.20 | 57.37 | 6354.78 | 1.43 | 5.24 |
| | 72 | 2.22 | 10.40 | 2.38 | 12.31 | 1.46 | 4.62 | 69.28 | 7990.50 | 3.42 | 27.88 |
| | 120 | 2.68 | 14.50 | 2.90 | 17.36 | 1.54 | 5.08 | 73.98 | 8594.37 | 4.40 | 43.76 |
| | 168 | 2.93 | 17.16 | 3.16 | 20.38 | 1.59 | 5.32 | 76.19 | 8860.52 | 4.92 | 52.77 |

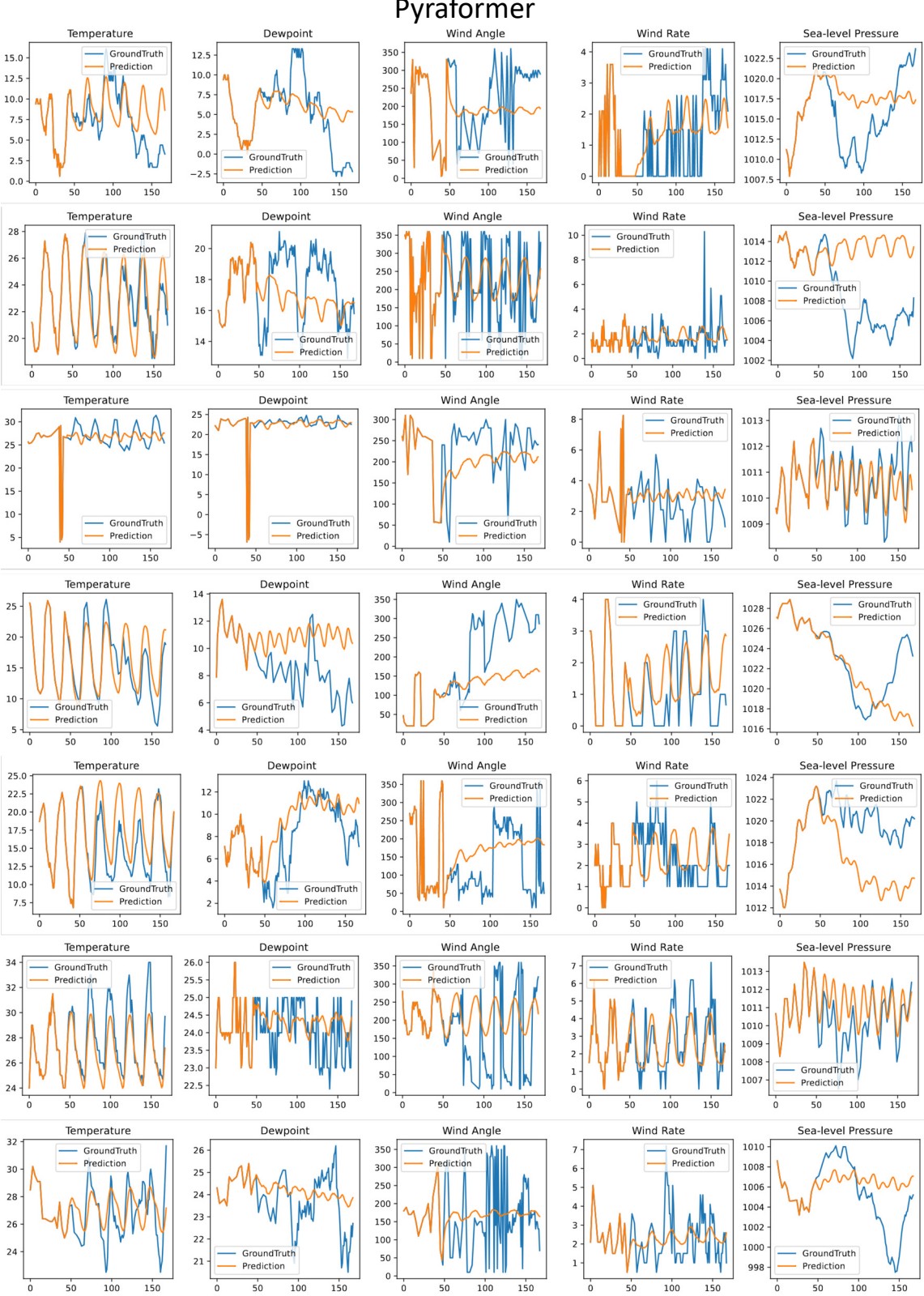

*Figure 10.* Visualization results of Pyraformer. Samples are randomly selected. Orange lines are predictions and Blue lines are ground truth.

# FEDformer

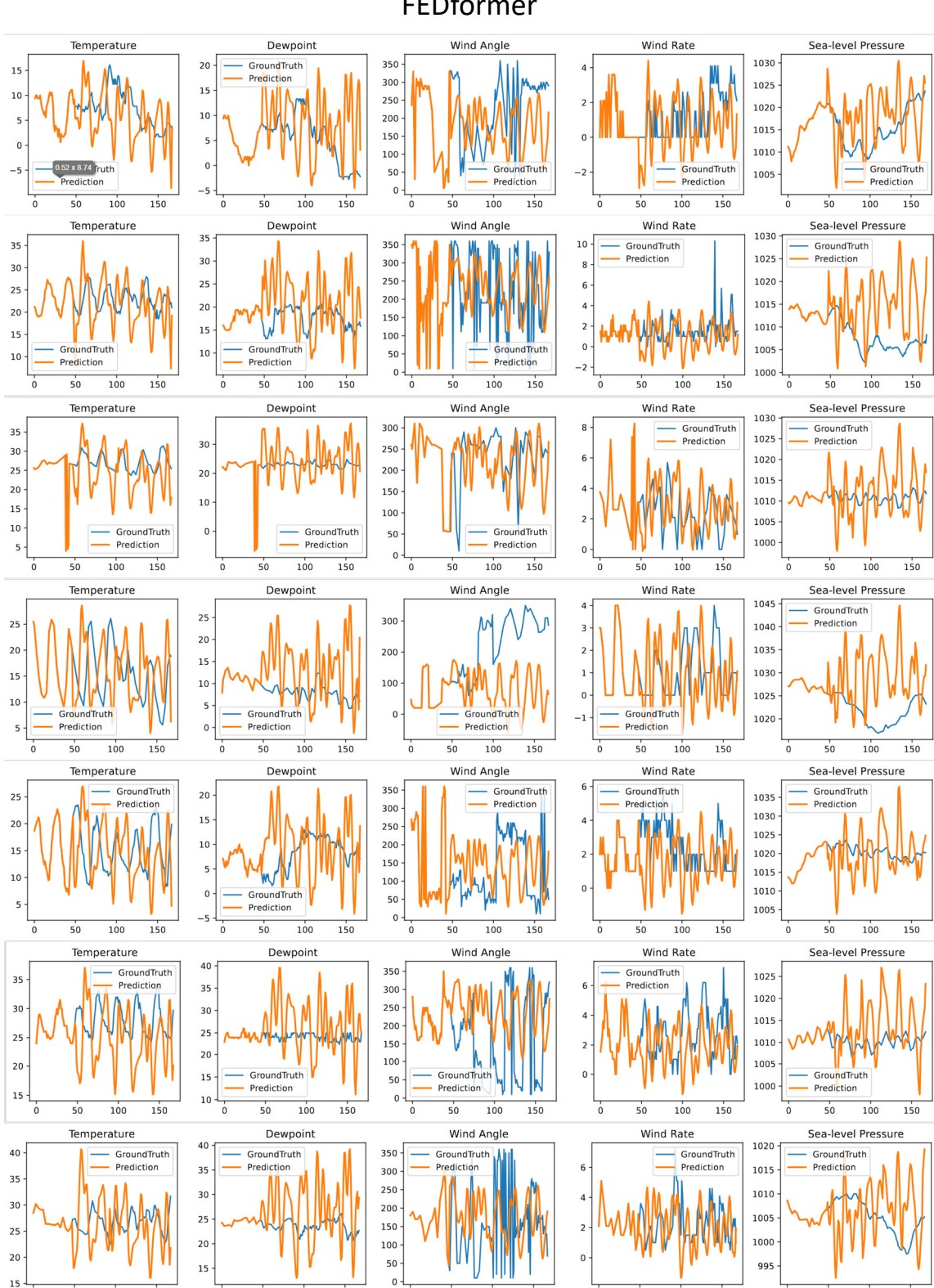

*Figure 11.* Visualization results of FEDformer. Samples are randomly selected. Orange lines are predictions and Blue lines are ground truth.

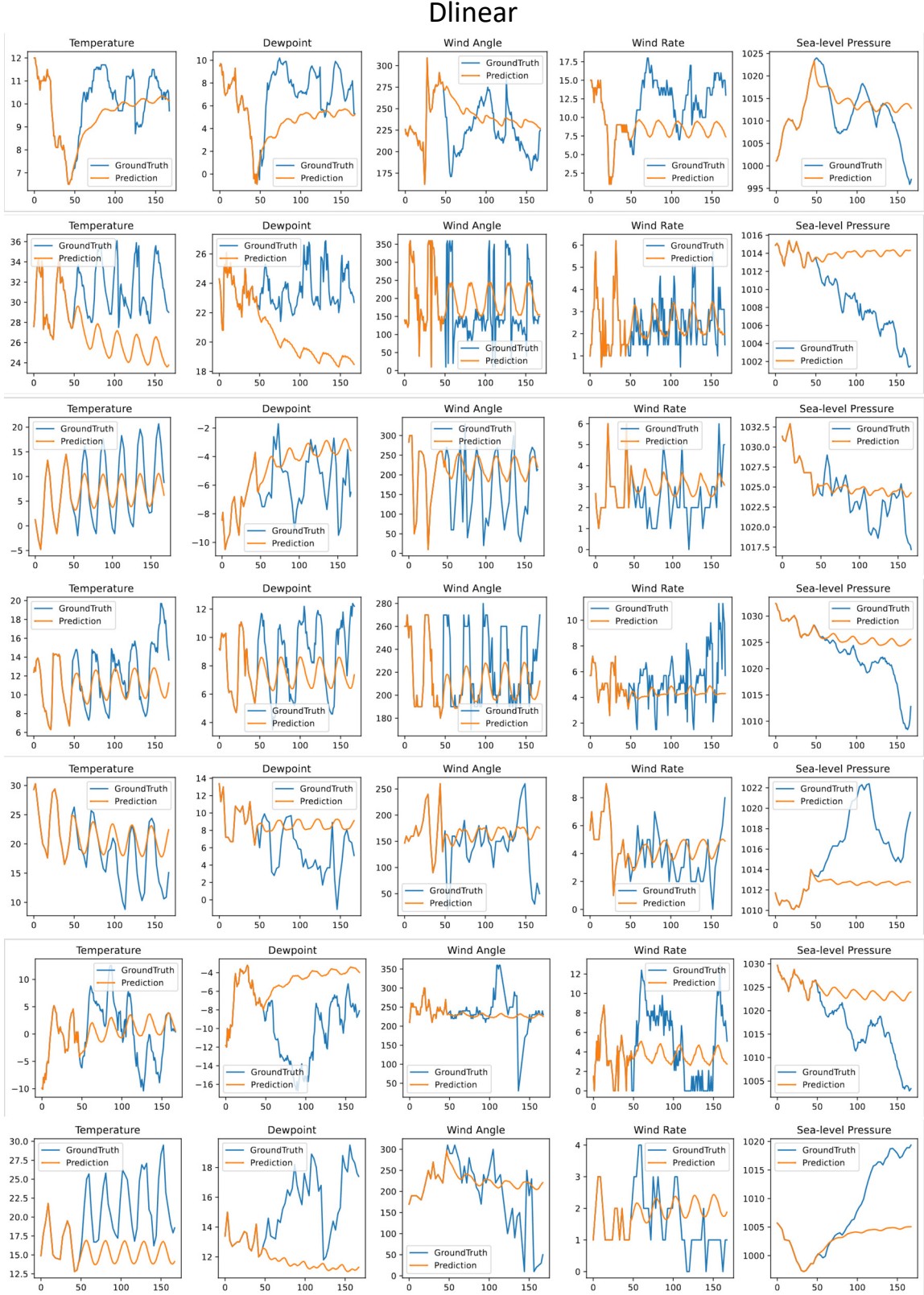

*Figure 12.* Visualization results of DLinear. Samples are randomly selected. Orange lines are predictions and Blue lines are ground truth.

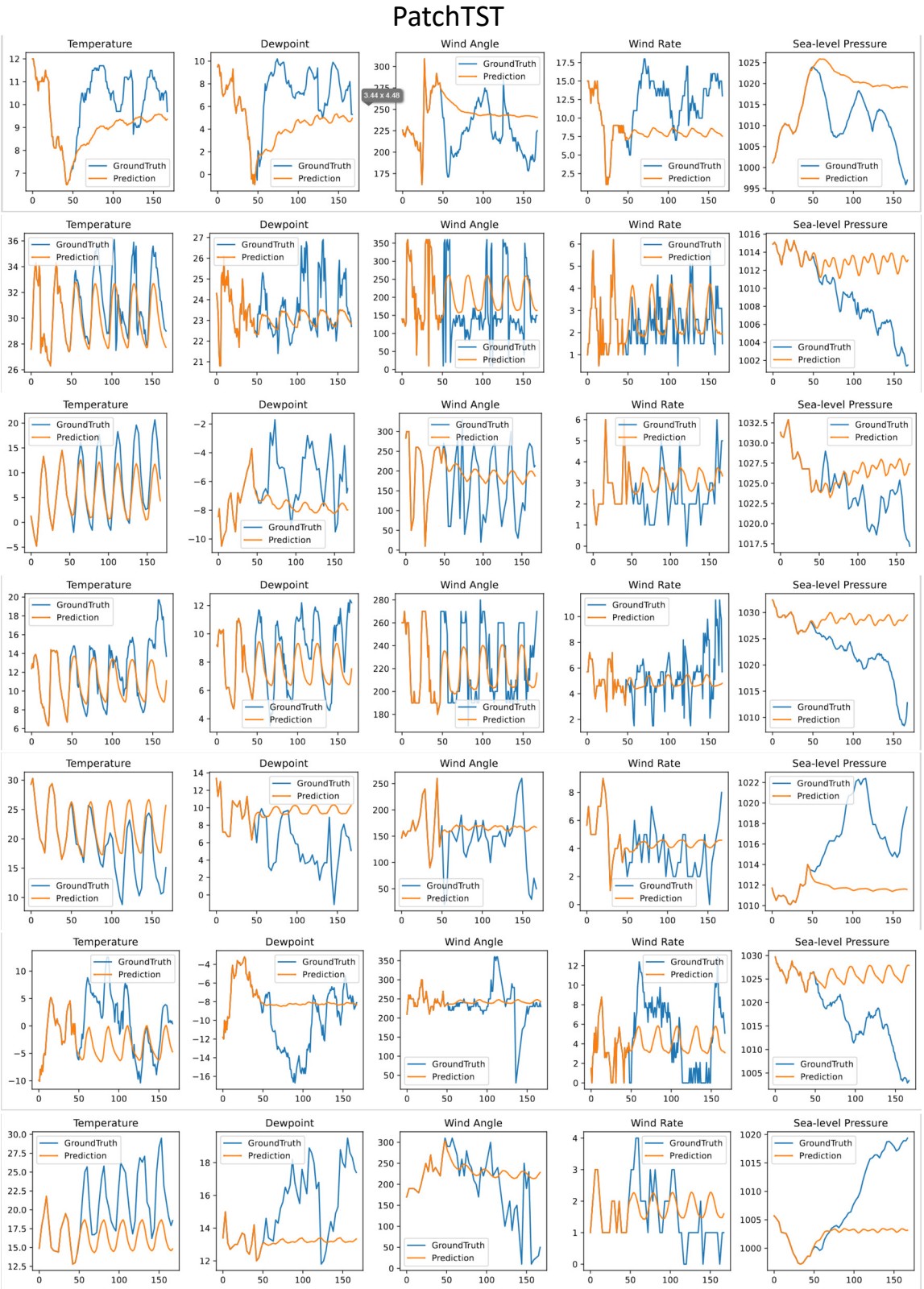

*Figure 13.* Visualization results of PatchTST. Samples are randomly selected. Orange lines are predictions and Blue lines are ground truth.

# iTransformer

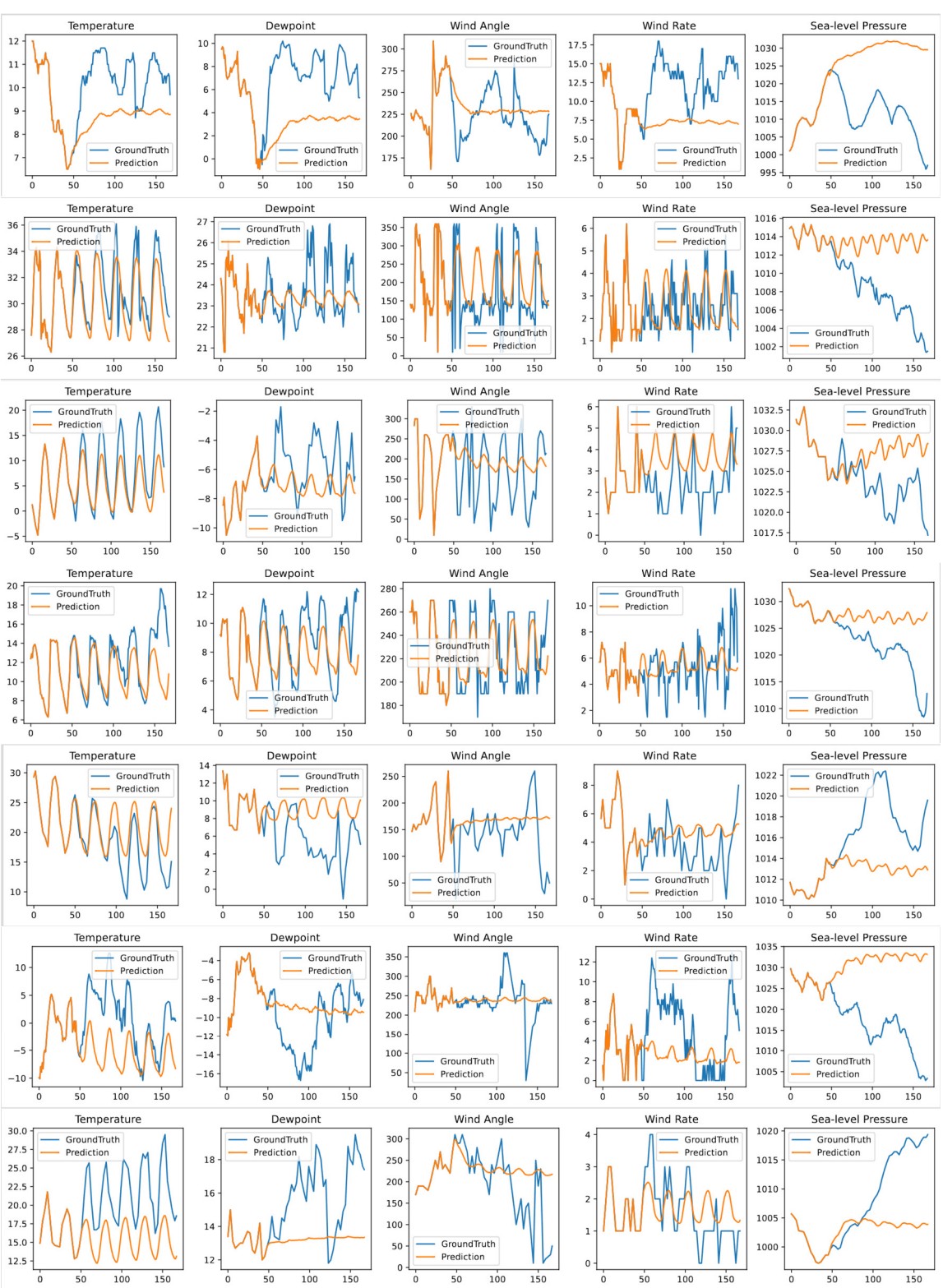

*Figure 14.* Visualization results of iTransformer. Samples are randomly selected. Orange lines are predictions and Blue lines are ground truth.

# Corrformer

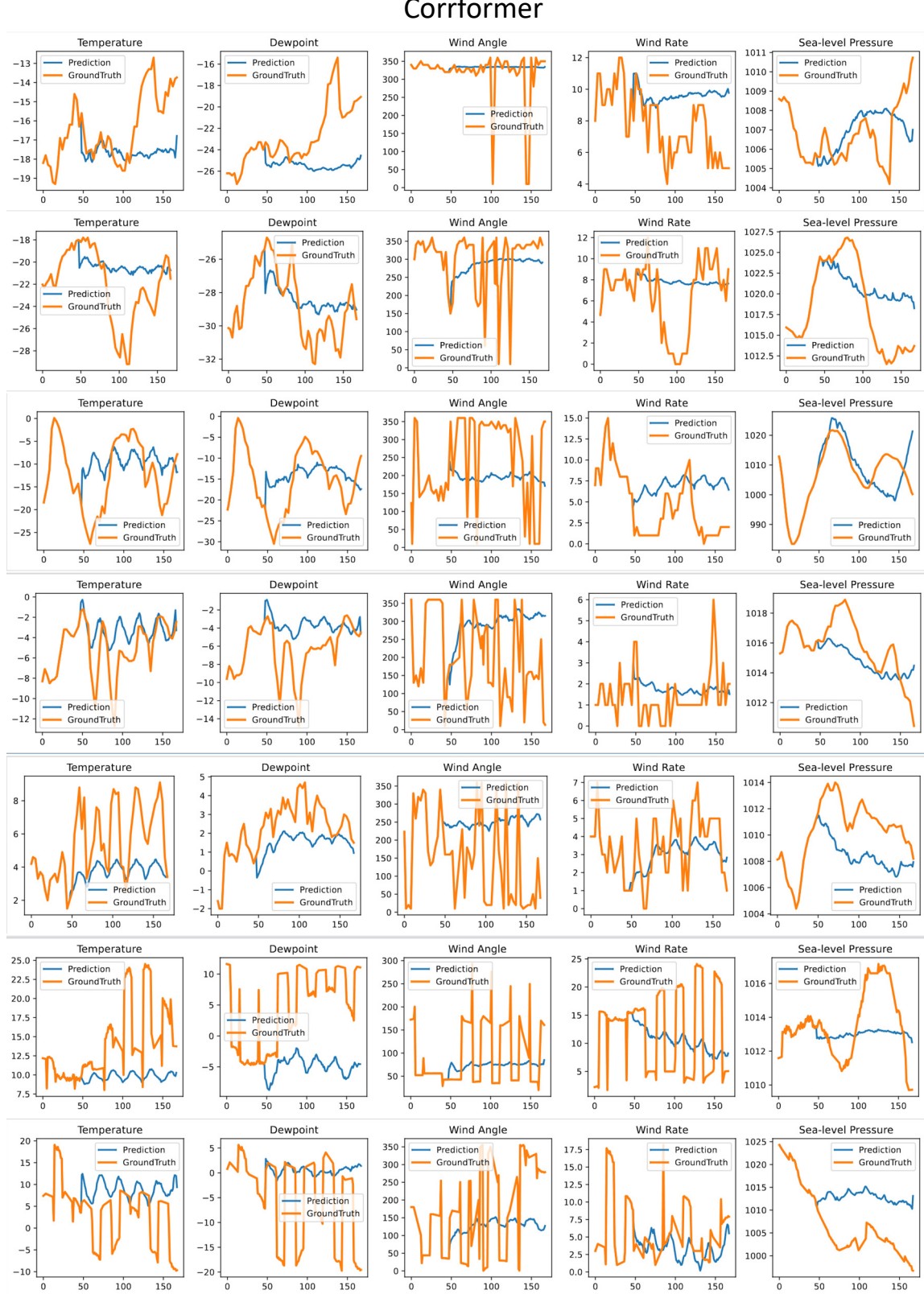

*Figure 15.* Visualization results of Corrformer. Samples are randomly selected. Orange lines are predictions and Blue lines are ground truth.

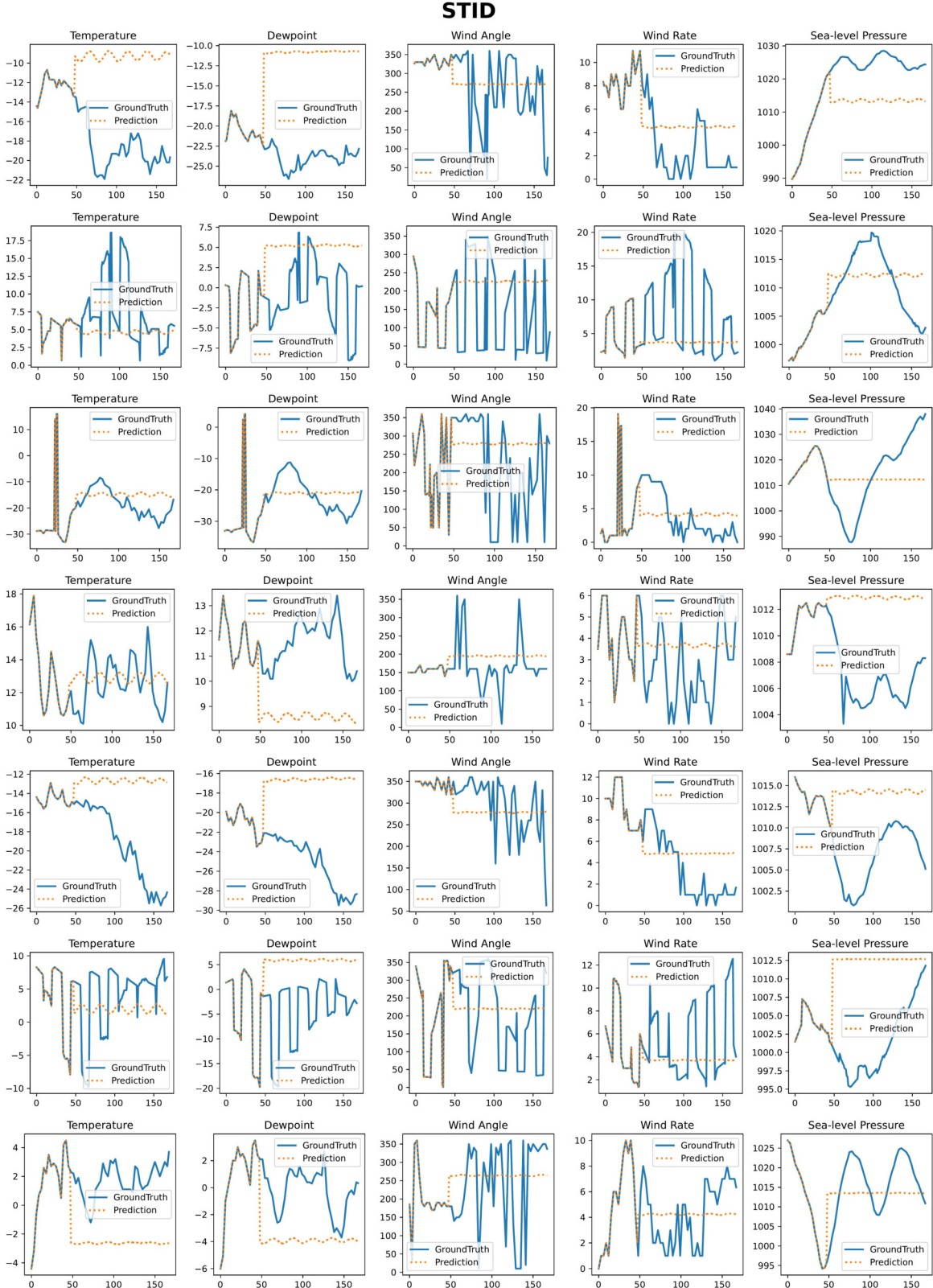

*Figure 16.* Visualization results of STID. Samples are randomly selected. Orange lines are predictions and Blue lines are ground truth.

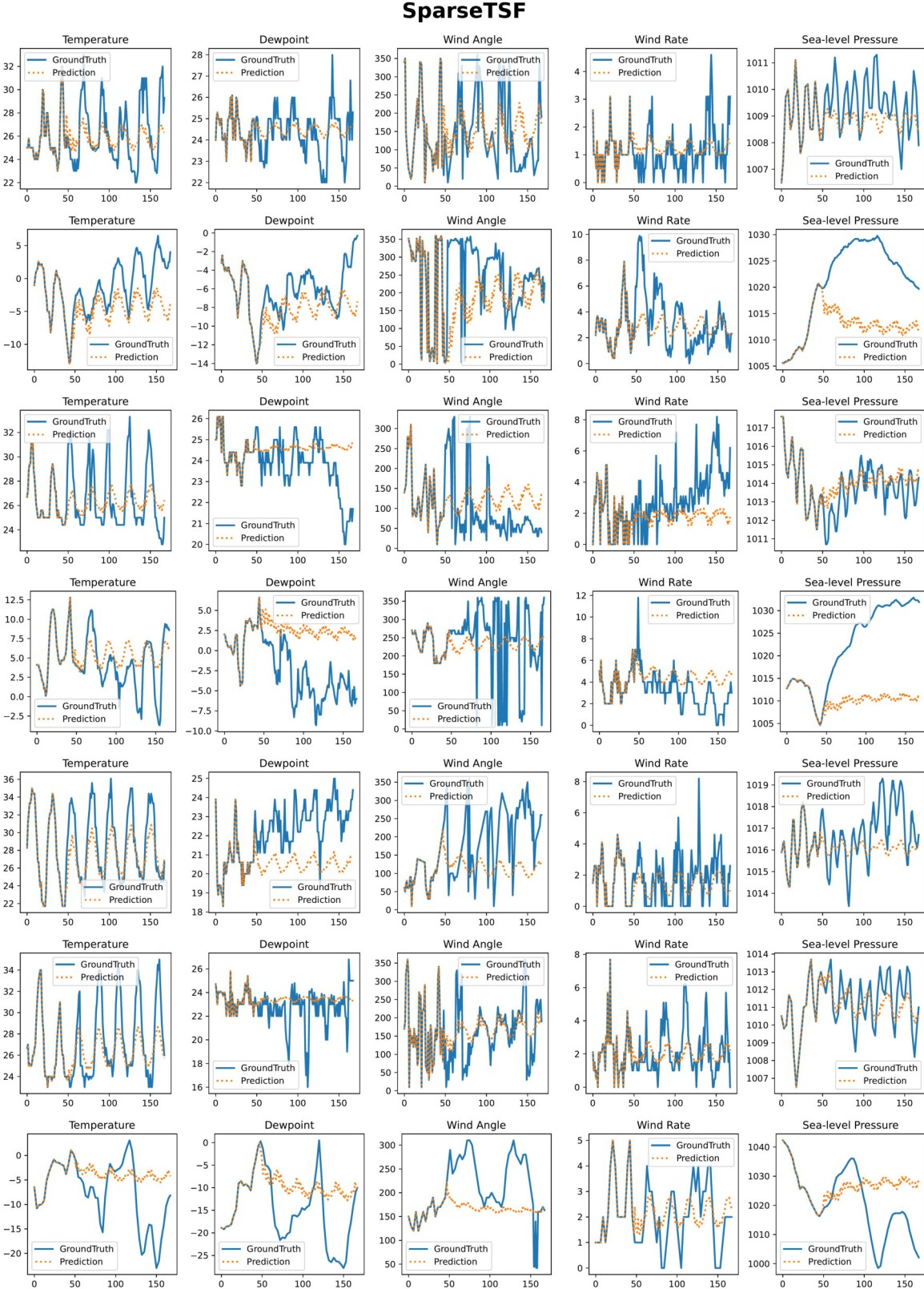

*Figure 17.* Visualization results of SparseTSF. Samples are randomly selected. Orange lines are predictions and Blue lines are ground truth.

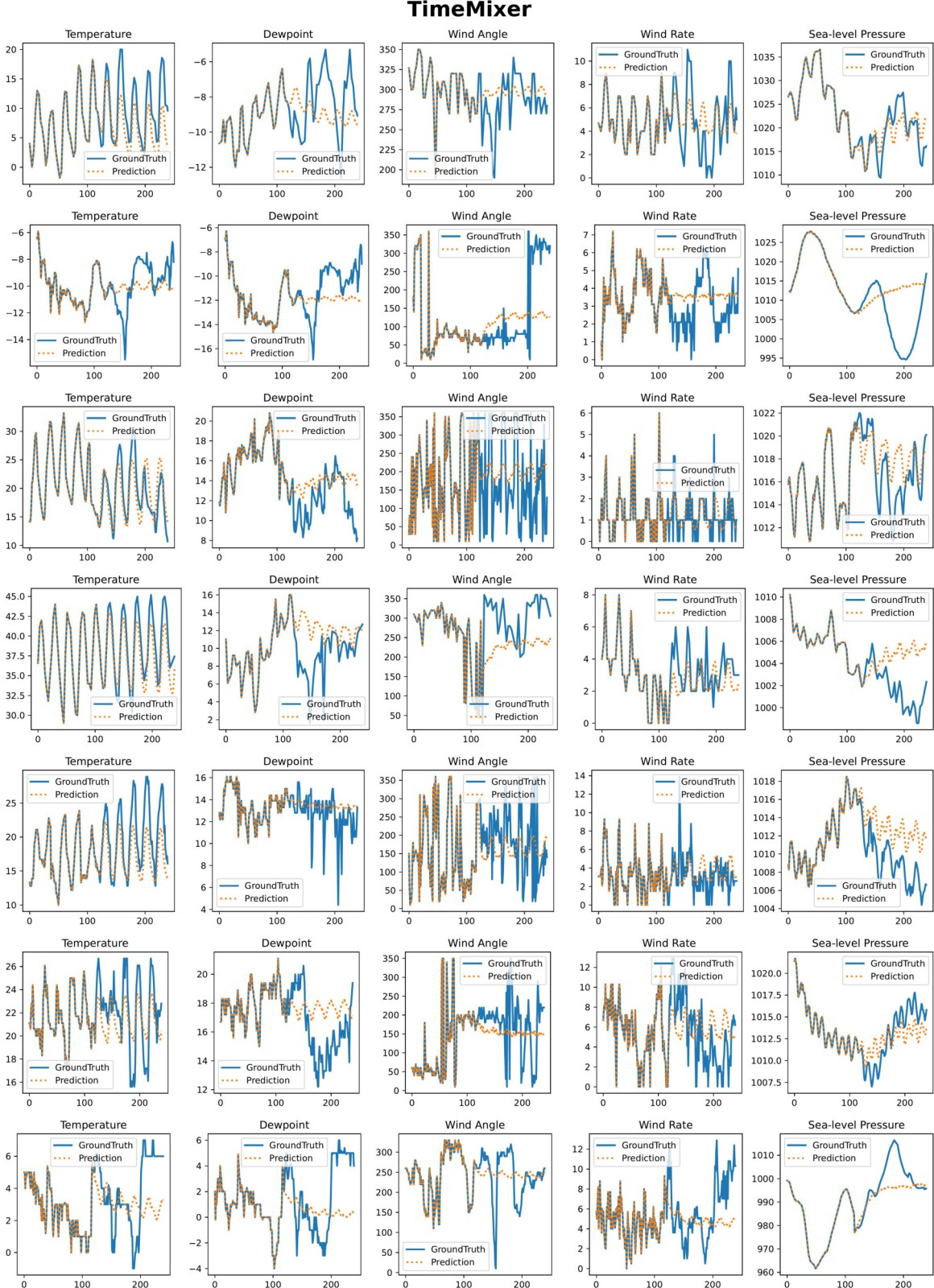

*Figure 18.* Visualization results of TimeMixer. Samples are randomly selected. Orange lines are predictions and Blue lines are ground truth.

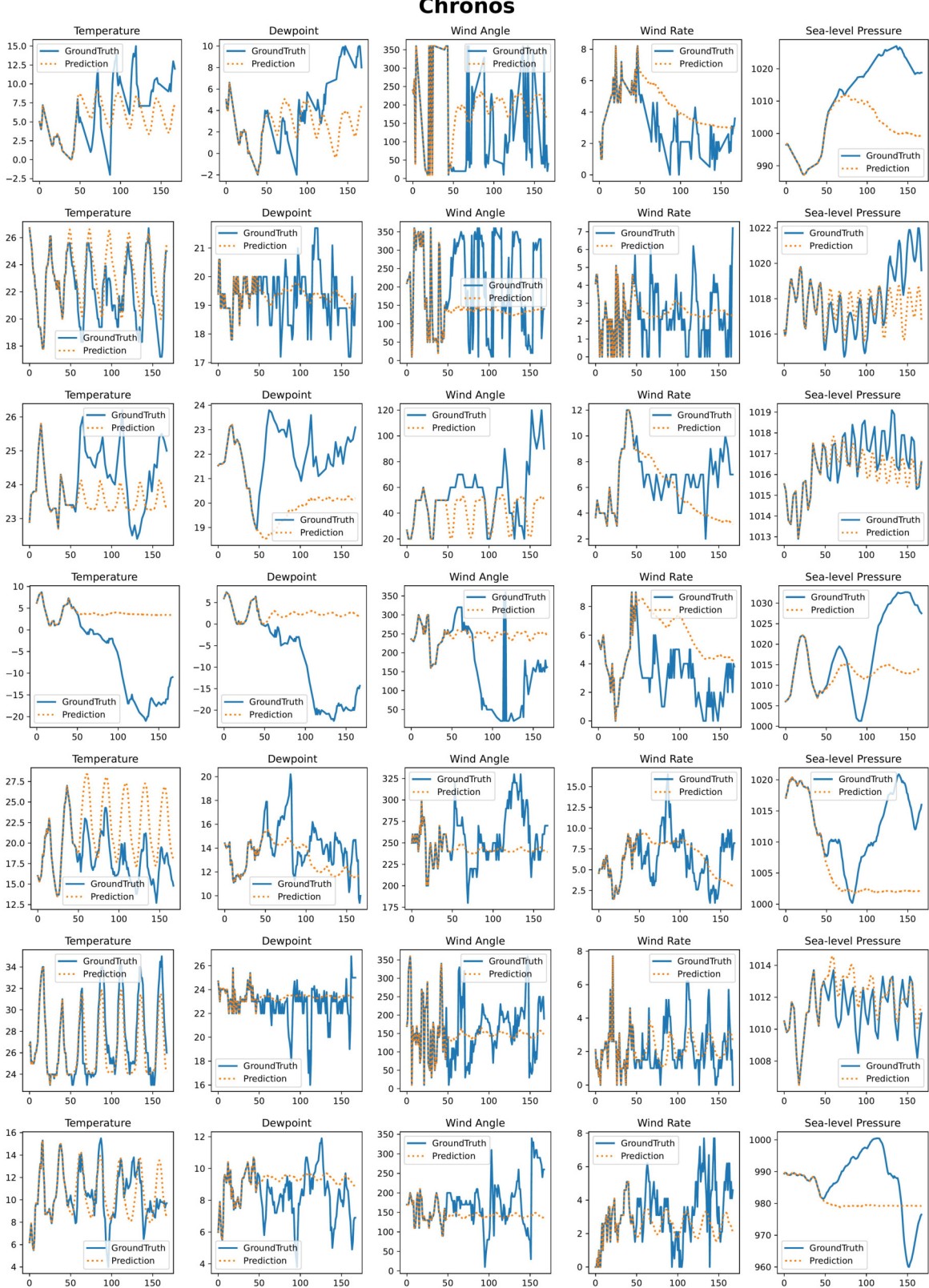

*Figure 19.* Visualization results of Chronos. Samples are randomly selected. Orange lines are predictions and Blue lines are ground truth.

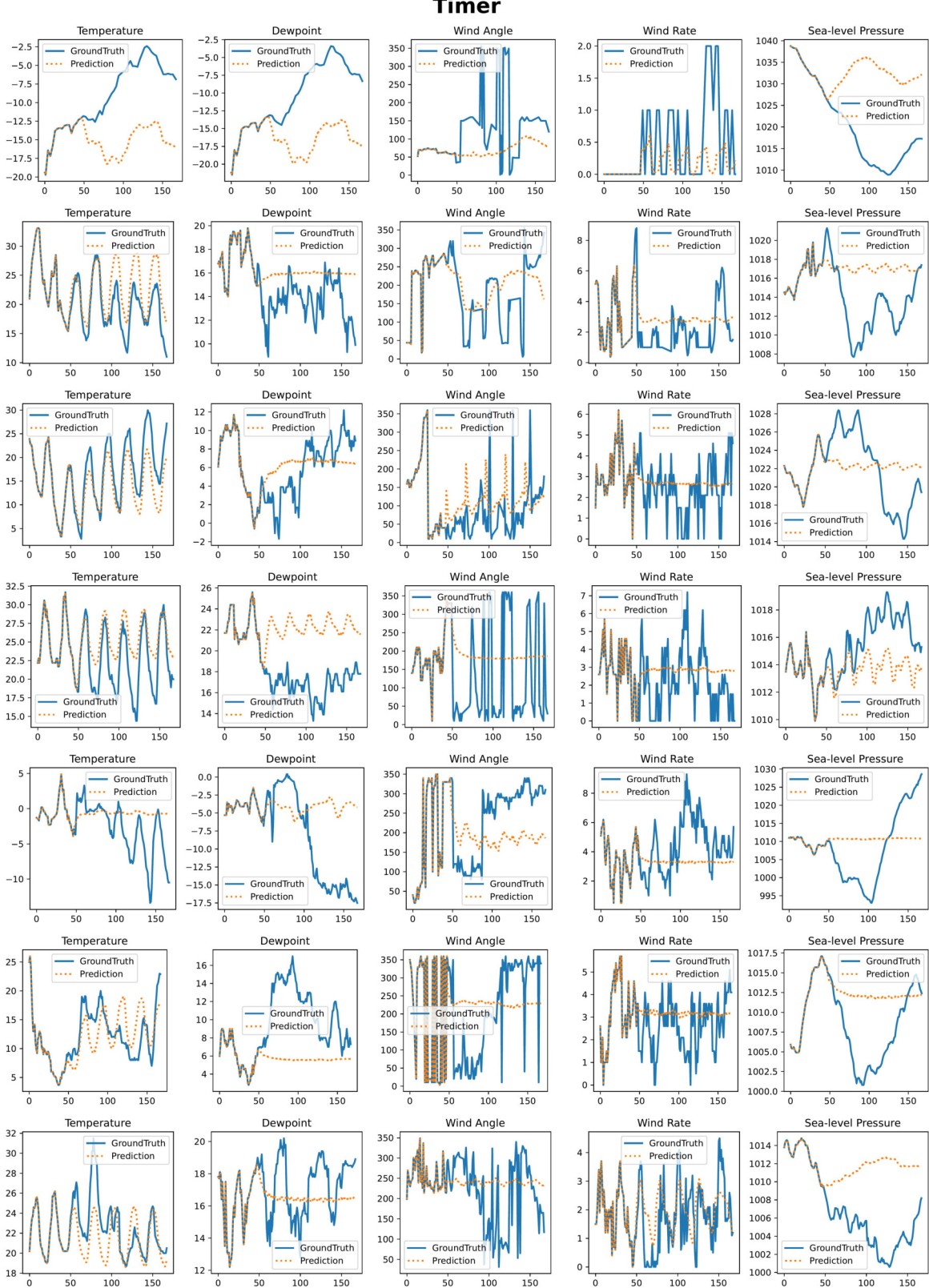

*Figure 20.* Visualization results of Timer. Samples are randomly selected. Orange lines are predictions and Blue lines are ground truth.

# PhysicsFormer (Ours)

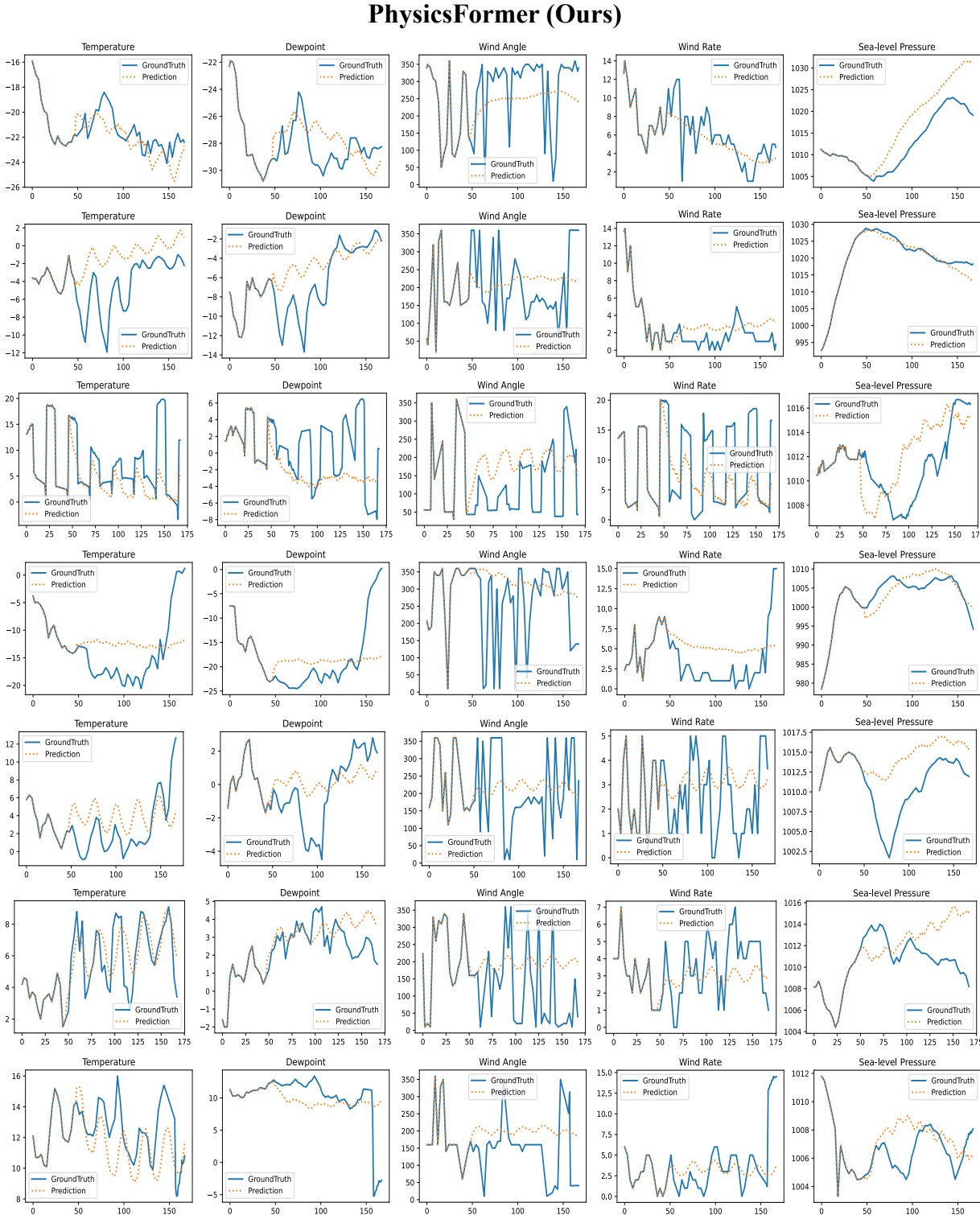

*Figure 21.* Visualization results of PhysicsFormer. Samples are randomly selected. Orange lines are predictions and Blue lines are ground truth.

