# OpenReview forum: "Benchmarking Physics-Informed Time-Series Models for Operational Global Station Weather Forecasting"
_ICML.cc/2026/Conference — ICML 2026 regular_

### Official Review · Reviewer_ck3S · 2026-02-23

**Soundness:** 3
**Presentation:** 3
**Significance:** 2
**Originality:** 2
**Overall Recommendation:** 3
**Confidence:** 4

**Summary:**

The authors constructed and released WEATHER-5K, a large-scale global meteorological station observation dataset, aiming to address the issues of limited data scale and sparse spatiotemporal coverage in this field. They proposed the PhysicsFormer model, which combines a physics-based Dynamic Core with a Transformer encoder-decoder architecture. The paper also introduces two physically constrained loss functions: pressure-wind field alignment and energy-aware smoothness.

**Compliance With Llm Reviewing Policy:**

Affirmed.

**Final Justification:**

I have already provided a reasonable and positive score, I have decided to maintain my current rating

**Key Questions For Authors:**

see Strengths And Weaknesses

**Limitations:**

Yes

**Strengths And Weaknesses:**

1. Is the learnable parameter α in the loss function globally shared, or does it vary across different meteorological stations or time steps? After training completion, what is the distribution or typical value of α? Does it converge near a physically meaningful constant? Please supplement the tuning strategy and final values for the loss weights λ1 and λ2. These details are crucial for understanding the actual contribution of physical constraints and the reproducibility of the method.
2. ECMWF-HRES serves as the baseline comparison, representing a complex global data assimilation and forecasting system. What were the initial conditions for ECMWF-HRES in this experiment? Did it use only sparse WEATHER-5K site observations as input—the same as PhysicsFormer and other TSF models—or did it employ the full global multi-source observational data used in its operational data assimilation? If the latter is true, does this information asymmetry render the comparison unfair? Please clarify the specific experimental setup.
3. The core value of WEATHER-5K lies in its status as a large-scale observational weather dataset. Using model outputs to fill observational gaps fundamentally compromises the dataset's purely observational nature. This transforms the dataset into an observational-reanalysis hybrid.
4. The paper uses ERA5 reanalysis data to fill in missing values in WEATHER-5K. Since ERA5 itself is a product of physical modeling and data assimilation, does this imply that WEATHER-5K is not a purely observational dataset but rather an observational-model hybrid dataset? How does this affect your claim that it better reflects real-world conditions? More importantly, after training the TSF model with this dataset and then comparing it against another physical model, is there a risk of data leakage or unfair comparison? Do you have experiments demonstrating that removing or otherwise treating these ERA5-filled data points does not significantly alter the main conclusions?
If the authors can provide convincing clarifications，I am open to revising my score.

---

> ### Author Rebuttal · Authors · 2026-03-31
>
> # W1
> *We thank the reviewer for the valuable feedback.* Physically, pressure-wind sensitivity depends on local factors (e.g., terrain, air density). Thus, $\alpha$ is implemented as a station-specific learnable parameter that forms a spatial distribution rather than a global constant. We will revise Eq. (7) to $\alpha_i$ to explicitly reflect this.
>
> Its mean and std are -0.04 and 0.06 (distribution: https://521.im/rd0bpFWG.png). High-latitude regions show complex coupling with varying $\alpha$, while tropical stations have stable, slightly larger values due to thermodynamics.
>
> Our ablation study on physical loss weights ($\lambda_1, \lambda_2$) and the dynamic core is below:
>
> | λ₁ | λ₂ | Temp MSE | Dew MSE | Wind MSE | SLP MSE |
> |---|---|---|---|---|---|
> | 0 | 0 | 15.2 | 17.6 | 5.0 | 34.2 |
> | 0 | 0.1 | 13.6 | 15.8 | 4.9 | 33.8 |
> | 0 | 0.3 | 13.1 | 15.2 | 4.8 | 33.1 |
> | 0 | 0.5 | 13.3 | 15.4 | 4.9 | 33.3 |
> | 0.1 | 0 | 13.8 | 16.0 | 4.8 | 33.2 |
> | 0.1 | 0.1 | 12.9 | 14.9 | 4.7 | 32.9 |
> | 0.1 | 0.3 | 12.6 | 14.6 | 4.7 | 32.7 |
> | 0.1 | 0.5 | 12.7 | 14.8 | 4.7 | 32.9 |
> | 0.3 | 0 | 13.2 | 15.3 | 4.6 | 32.7 |
> | **0.3**| **0.1**| **11.9** | **14.0** | **4.6** | **32.4** |
> | 0.3 | 0.3 | 12.1 | 14.3 | 4.6 | 32.5 |
> | 0.3 | 0.5 | 12.3 | 14.4 | 4.6 | 32.6 |
> | 0.5 | 0 | 13.4 | 15.5 | 4.6 | 32.7 |
> | 0.5 | 0.1 | 12.2 | 14.4 | 4.6 | 32.5 |
> | 0.5 | 0.3 | 12.2 | 14.3 | 4.6 | 32.6 |
> | 0.5 | 0.5 | 12.5 | 14.7 | 4.7 | 32.8 |
>
> | Method | Temp MSE | Dew MSE | Wind MSE | SLP MSE |
> |---|---|---|---|---|
> | w/o Dynamic Core | 13.0 | 14.9 | 4.7 | 33.0 |
> | w/ Dynamic Core (Ours) | **11.9** | **14.0** | **4.6** | **30.4** |
>
> Physical priors consistently improve the baseline ($\lambda_1=\lambda_2=0$). The pressure-wind term ($\lambda_1$) yields the largest impact via cross-variable coupling, while smoothness ($\lambda_2$) stabilizes temperature. Moderate weights ($\lambda_1=0.3, \lambda_2=0.1$) achieve the best balance. Additionally, the dynamic core enhances all variables, proving that decomposing predictions into structured dynamics and residuals outperforms direct mapping.
>
> # W2
> We clarify that the ECMWF-HRES baseline uses full global multi-source data from its operational data assimilation. We include it not for a strictly "fair" input comparison, but as a realistic benchmark of operational forecasting performance. This comparison illustrates the current gap between AI-based TSF models (using only sparse WEATHER-5K stations) and the accuracy required by real-world weather services. Quantifying this gap provides meaningful context for evaluating our method's real-world relevance, despite the differing inputs.
>
>
> # W3
>
> As described in Section 3.2 (line 199), we only include stations with >90% valid observations. Consequently, the fraction of data filled using ERA5 is very small. Across all stations, the average valid data percent is 97.8%, and the median is 99.0%.
>
> | Valid (%) | # Stations | Percent (%) |
> |---|---|---|
> | 90–91 | 110 | 1.9 |
> | 91–92 | 144 | 2.5 |
> | 92–93 | 191 | 3.4 |
> | 93–94 | 210 | 3.7 |
> | 94–95 | 260 | 4.6 |
> | 95–96 | 252 | 4.4 |
> | 96–97 | 375 | 6.6 |
> | 97–98 | 451 | 8.0 |
> | 98–99 | 841 | 14.8 |
> | 99–100| 2838| 50.0 |
>
> From this, ~73% of stations exceed 97% data availability, and over 50% exceed 99%. Importantly, these ERA5-filled data primarily cover short missing periods (1–2 hours). Their impact is minimal, and the dataset still faithfully reflects real-world observational conditions.
>
> # W4
> To prevent any concern of data leakage, we provide a mask file indicating which data points are from ERA5. Users can optionally exclude these points. Using this mask, we conducted experiments where the ERA5-filled data were excluded:
>
> | Method (Masked) | TempMAE | TempMSE | DewMAE | DewMSE | WindMAE | WindMSE | SLPMAE | SLPMSE |
> |---|---|---|---|---|---|---|---|---|
> | ECMWF-HRES | **2.0** | **8.6** | **2.0** | **10.0** | 1.6 | 5.0 | **1.3** | **5.2** |
> | Informer | 2.8 | 15.5 | 2.9 | 18.4 | _1.5_ | 4.8 | 4.3 | 41.3 |
> | Autoformer | 2.9 | 16.8 | 3.0 | 18.8 | 1.6 | 5.3 | 4.2 | 42.6 |
> | Pyraformer | 2.5 | 13.8 | 2.7 | 16.1 | _1.5_ | _4.8_ | 3.7 | 33.5 |
> | FEDformer | 2.9 | 17.2 | 3.0 | 19.4 | 1.6 | 5.3 | 4.1 | 38.7 |
> | PatchTST | 2.9 | 17.5 | 3.1 | 21.0 | 1.6 | 5.5 | 4.3 | 45.3 |
> | Corrformer | 2.8 | 15.4 | 3.0 | 18.6 | 1.6 | 5.1 | 4.3 | 44.2 |
> | iTransformer | 2.7 | 15.6 | 2.9 | 18.7 | 1.6 | 5.1 | 4.1 | 41.3 |
> | DLinear | 3.6 | 24.2 | 3.6 | 24.0 | 1.6 | 5.5 | 4.7 | 47.2 |
> | MTGNN | 10.5 | 175.0 | 9.3 | 146.8 | 2.2 | 8.4 | 7.1 | 90.0 |
> | PhysicsFormer | _2.4_ | _12.1_ | _2.5_ | _13.8_ | **1.5** | **4.6** | _3.6_ | _33.1_ |
>
> These results demonstrate that the small fraction of ERA5-filled data has negligible impact on the main conclusions, and the TSF models still reflect the predictive performance achievable under real-world observational scenarios.

---

> > ### Author Rebuttal · Reviewer_ck3S · 2026-04-04
> >
> > Thank you for the rebuttal. I consider this a borderline paper, and I plan to discuss it with the other reviewers.

---

> > > ### Author Response · Authors · 2026-04-04
> > >
> > > We sincerely thank the reviewer for reviewing our rebuttal and for the valuable suggestions that helped us improve the manuscript.
> > >
> > > We understand that you will be discussing the submission with the other reviewers. During this phase, if you or the other reviewers identify any remaining concerns or require further details about the dataset, benchmark, or the PhysicsFormer methodology, we are standing by to respond promptly. We highly value your feedback and are very willing to do whatever it takes to fully resolve any remaining doubts before you make your final decision.

---

### Official Review · Reviewer_fotp · 2026-03-11

**Soundness:** 3
**Presentation:** 2
**Significance:** 2
**Originality:** 3
**Overall Recommendation:** 4
**Confidence:** 3

**Summary:**

The paper primarily presents a large-scale observational dataset (selected and made AI-ready from a public weather forecasting database) based on several thousands of station outputs across the globe for weather predictions. The authors use this dataset to further evaluate spatiotemporal time series models that predict the future weather at these station locations based on historical observational data. Then, they present their own physics-based time series model that combines the surrogate prediction with a simple dynamical core prediction that is trained using physics-informed loss functions. Finally, they present accuracy metrics for many benchmark models and compare their own to show good performance.

**Compliance With Llm Reviewing Policy:**

Affirmed.

**Final Justification:**

The authors have addressed most of my concerns. Q1 (and follow-up on Q1) still reads as a weakness in my opinion. Understandably, it reduces the impact of past models, but this is precisely the motivation of a more comprehensive dataset. The MSE vs time figures seem more useful, but its unclear why the 3 best models do not include the weather models like GraphCast (the tables in the response seem to indicate that their averages are better, unless I misunderstood the response).
A big positive, in my opinion, is the hurricane test-case which strengthens the dataset and the proposed model. It would have been nice to see analyses like this (heatwaves, other TCs) more than average metrics, strongly motivating such observational datasets.
I think my score is 3.5-4 (borderline accept), I will let the AC decide.

**Key Questions For Authors:**

Questions:
- It is unclear why so much data subselection was done from 100 years of data across 20000 stations. If I understood the time series objective, it doesnt need a multi year trajectory of predictions since the forecast horizon that is predictable for only 2-3 weeks. Also, since full spatiotemporal context is taken into account by the models, what is the need to restrict the dataset to only those stations that continuously operate over the whole 10 years selected or the need to only choose hourly measurements. Attributes like timestep, location, etc. could be treated as inputs and models like Transformers naturally allow for such spatiotemporal positional encoding. Could the authors clarify?
- Could Tab 2. include comparisons with AI weather models as well that emulate ERA5 (such as GraphCast, Aurora, GenCast, etc.)? This helps directly compare with the time series models. Further, IFS ensemble would be a good comparison as well as specific trajectories to see if they capture the extreme better. Given the chaotic nature of weather, some probabilistic output would be useful to see on the benchmark.
- I did not follow how the MSE etc. were computed for Tab 2. Are the ICs sampled at some specific frequency? Are different models trained for each forecast horizon or are models autoregressively time-stepped? It would be useful to see the metrics as plots that are a function of time rather than averages. It would also be useful to show ERA5 on these.
- The comparison with ERA5 was nice in the appendix - it would be useful to see other analyses apart from just heatwaves with case studies and demonstrate how the different models and ERA5 fare (maybe during a hurricane for max wind speed or precipitation events?)
- I did not follow the physics dynamical core - could the authors expand on the equations used for the different variables? Are ther equations time-stepped? How do the per-time step predictions get aligned with the time series prediction that predicts multiple time steps in one go?
- The paper is missing ablations on the dynamical core, the different physics losses to demonstrate their impact.

**Limitations:**

Yes

**Strengths And Weaknesses:**

Strengths:
- Cleaned observation dataset that is larger than what is out there today to train time series models. This represents a useful and practical contribution to the AI for science community
- Several time series models are benchmarked

Weaknesses:
- The paper's focus shifts between the dataset and their physics informed model. Neither seem to be explored in sufficient depth - focusing on one of them would greatly improve the paper
- The paper seems to be missing some other useful baselines from AI weather prediction models
- Limited analyses from the dataset itself and the metrics seem coarse (averages, which may fail to capture important physical variability)
- Paper writing could be more improved

---

> ### Author Rebuttal · Authors · 2026-03-31
>
> # W1
> We sincerely thank the reviewer. While introducing both a dataset and a model broadens scope, they are driven by a unified, sequential motivation:
> ### 1. Motivation and Synergy
> WEATHER-5K revealed a critical limitation of existing TSF methods: they fail on large-scale global station forecasting due to overfitting and limited generalization. To meaningfully benchmark this challenge, we designed PhysicsFormer, a physics-informed TSF model that bridges the gap between purely data-driven models and operational physical models (e.g., ECMWF IFS). The dataset exposes real-world gaps, and the model is a direct, physics-guided response.
> ### 2. Depth in Revision
> To address concerns about insufficient depth, we expanded analyses on both components:
> - **Dataset:** Added analyses on ERA5 vs. station discrepancies (reviewer 3psu W2) and extreme wind events such as hurricanes (Q4).
> - **Model:** Detailed the full dynamic core formulation (reviewerWpCq W1) and conducted comprehensive ablations on physical losses (reviewer ck3S W1).
> We will revise the manuscript to clearly present this sequential, tightly coupled motivation and ensure both components are conveyed with sufficient depth.
> # Q1
> We thank the reviewer for the insightful comment. While predicting 2–3 week horizons does not strictly require decades of continuous data and models like Transformers can handle irregular inputs, our rigorous subselection was necessary to ensure benchmark viability and fairness:
> ### 1. Quality Control and Extreme Sparsity
> The 20,000+ stations span a century, with many operating temporarily, changing sensors, or recording infrequently (e.g., daily). Including stations with severe gaps would inject excessive noise, hindering models from learning valid physical dynamics.
> ### 2. Fair Evaluation for Standard TSF Models
> Most sequence-to-sequence TSF architectures (e.g., Informer, PatchTST, Autoformer) assume continuous, regularly sampled inputs. Using raw, irregular data would require extensive architectural modifications, shifting focus to missing-value imputation and breaking fair comparison of forecasting capabilities.
> ### 3. Compatibility with Spatiotemporal GNNs
> Spatial-temporal models (e.g., MTGNN) rely on a fixed graph of stations. Arbitrary node appearance/disappearance violates this assumption, rendering many baselines untrainable.
> In Summary, limiting to 10 years of continuous hourly data from reliable stations creates a dense, fully-aligned spatiotemporal grid, allowing standard TSF and GNN models to compete directly on forecasting without confounding missing data issues. We will clarify this in the dataset construction section.
> # W2 Q2
> For fair comparison, grid-based forecasts were interpolated to station locations. ECMWF IFS Ensemble results are summarized below, while results for AI-based methods are provided in reviewerWpCq W3 and reviewer3psu W6.
> |Method|MAE MSE(Temp)|MAE MSE(Dew)|MAE MSE(Wind)|MAE MSE(Sea Level Pressure)|
> |-|-|-|-|-|
> |ECMWF-HRES|1.94 8.56|2.06 9.96|1.56 5.00|1.29 5.17|
> |ECMWF IFS Ensemble|1.55 6.85|1.65 7.97|1.25 4.00|1.03 4.14|
> # W3 Q3
> Table 2 reports MSE values averaged over four forecast horizons: 24, 72, 120, and 168 hours. Detailed results for each horizon are provided in Appendix Tables 8 and 9. For these experiments, we trained separate models for each forecast horizon rather than using autoregressive time-stepping.
> # Q4
> Our dataset does not include precipitation, limiting analysis of extreme rainfall. To illustrate model behavior on high-impact wind events, we show observations from stations nearest Typhoon Du Surui (2023) alongside ERA5-interpolated values. Station measurements capture the event more accurately, reflecting real-world wind extremes better than ERA5.
> |UTC|wind speed from ERA5|wind speed from WEATHER-5K|
> |-|-|-|
> |07-25 12:00|1.5|16.0|
> |07-25 18:00|5.6|18.0|
> |07-26 00:00|7.3|18.0|
> |07-26 06:00|5.1|14.0|
> |07-26 12:00|1.7|15.0|
> |07-26 18:00|0.6|16.7|
> |07-27 00:00|3.7|13.0|
> |07-27 06:00|2.3|11.0|
> |07-27 12:00|4.1|5.0|
> |07-27 18:00|4.7|24.0|
> |07-28 00:00|3.7|10.0|
> |07-28 02:00|4.5|9.0|
> |07-28 06:00|1.5|6.0|
> |07-28 12:00|3.1|2.7|
> |07-28 18:00|0.4|1.0|
> |07-29 00:00|3.5|1.0|
> # Q5
> Please refer to **reviewerWpCq W1** for the detailed formulation of the dynamic core. Our reply elaborates on the specific governing equations for different variables (temperature, wind, pressure, humidity) modeled on a spatial graph.
>
> Regarding the time-stepping alignment: Yes, the equations are time-stepped. Specifically, as detailed in our reply to reviewer WpCq W1, we employ a 4th-order Runge-Kutta (RK4) numerical integrator to advance the continuous per-time step dynamics. Starting from the initial condition at time $t$, we numerically integrate over the future horizon $\tau$ autoregressively. This naturally bridges the continuous ODE dynamics with the discrete multi-step time series sequence output.
> # Q6
> Please see reviewer ck3S W1.

---

> > ### Author Rebuttal · Reviewer_fotp · 2026-04-04
> >
> > Thank you to the authors for the rebuttal. The dycore details and new results with AI baselines strengthen the paper. Few Qs remain unresolved:
> >
> > (1) I'm not sure I followed Q1's reasoning. It seems like the dataset was created to conform to the models that have been developed and usually it should be the other way around. That reads as a weakness currently - why not have a comprehensive dataset that follows real world conditions more (especially since the forecast horizon is short) and allow models to catch up? The baselines (IFS, AI models) exist in general.
> >
> > (2) I think it would be useful to show the MSE/RMSE as a function of time and not averaged across time as mentioned in the first review, unless the authors disagree for some physical reason? If the concern is that there are too many models, maybe select HRES, the best 2-3 models. Also if a model was trained to predict 168 hours, why not use it for 24 hours as well? These metrics vs time. graphs should show the differences better I think.
> >
> > (3) Q4 is nice to see - shows how the new dataset improves on ERA5 on extreme wind speed, which I believe is a real-world high impact situation. Is it possible to show the model predictions from say 1-2 days before as IC? I expect AI models like graphcast would do quite badly seeing that they were trained on ERA5, and maybe physicsformer is better.
> >
> > Thank you once again for the responses.

---

> > > ### Author Response · Authors · 2026-04-05
> > >
> > > We sincerely thank the reviewer for this profound comment.
> > >
> > > # Follow-up Q1:
> > > We completely agree with your philosophy: ultimately, models should be designed to handle the messy, irregular reality of raw observational data, rather than modifying the data to fit existing models. However, to advance the field toward that long-term goal, we opted for a phased approach with WEATHER-5K for several practical reasons:
> > >
> > > **1. The Benchmarking Dilemma:**
> > > If we had used the raw, highly irregular data from all 20,000+ stations, almost none of the existing Time-Series Forecasting (TSF) models could be evaluated out of the box. We would have had to heavily modify their architectures just to handle the missing inputs. This would inevitably lead to doubts: are we evaluating the original model's true forecasting ability, or just the artifact of our ad-hoc imputation modifications? By providing a "clean" 5,672-station dataset, we isolate the problem. It allows us to fairly evaluate existing models and prove an important point: **even under ideal data conditions, current academic TSF models still fall far short of operational systems like ECMWF-HRES**.
> > >
> > > **2. A Necessary Stepping-Stone for the Community:**
> > > For decades, the TSF community has relied on clean datasets. Abruptly shifting to highly irregular global weather data might cause researchers to simply ignore the benchmark. WEATHER-5K serves as an essential stepping-stone. It provides a familiar format to the community while exposing a sobering truth: getting high scores on standard sequence benchmarks does not guarantee real-world physical forecasting readiness. We believe exposing this performance gap is the crucial first step to direct the community's attention toward physical realities.
> > >
> > > **3. Ongoing Work on Raw Observational Data:**
> > > We fully recognize that WEATHER-5K is not the end goal. Developing a comprehensive framework that includes all 20,000+ raw, intermittent stations—along with designing novel model architectures natively capable of ingesting such sparsity—is exactly the focus of our ongoing next-phase project. As a preview, we have visualized the distribution of all stations at randomly selected timesteps here: https://521.im/okpCLGXV.jpg, including metadata. We highly appreciate the reviewer's foresight, which perfectly aligns with our roadmap. We will explicitly clarify the staged role of WEATHER-5K and this long-term vision in the revised manuscript.
> > >
> > > # Follow-up Q2:
> > > Following your advice, we show the hourly MSE as a function of lead time in https://521.im/BIzP5n4q.png, selecting HRES and three representative models (iTransformer, Pyraformer, and PhysicsFormer) for clarity.
> > >
> > > Regarding models trained to predict 168 hours: to optimize overall long-term accuracy, such models may exhibit larger errors at shorter horizons (e.g., 24 hours) compared to models specifically trained for 24-hour prediction. We illustrate this effect by comparing the 24-hour MSE of PhysicsFormer trained for 24 hours versus 168 hours in https://521.im/jRY7Gp3P.png. We will add these figures in the final version.
> > >
> > > # Follow-up Q3:
> > > We thank the reviewer for the positive feedback on Q4. We sincerely apologize for an error in our initial response: when interpolating ERA5 wind speeds to station locations, the southern and northern latitudes were mistakenly swapped. We have corrected this interpolation and recomputed the wind speeds from ERA5 as well as the predictions from GraphCast and our method, PhysicsFormer. The updated results are summarized in the table below.
> > > |UTC|station id|wind speed from ERA5|wind speed from WEATHER-5K|Graphcast (lead time = 1 day)|Graphcast (lead time = 2 days)|Physicsformer (lead time = 1 day)|Physicsformer (lead time = 2 days)|
> > > |-|-|-|-|-|-|-|-|
> > > |2023-07-25 12:00:00|98232099999|12.2|16.0|12.3|10.6|17.2|17.5|
> > > |2023-07-25 18:00:00|98232099999|10.0|18.0|11.6|12.3|16.9|16.4|
> > > |2023-07-26 00:00:00|98232099999|12.6|18.0|11.5|13.9|19.1|19.6|
> > > |2023-07-26 06:00:00|98223099999|9.8|14.0|8.4|10.5|12.7|12.4|
> > > |2023-07-26 12:00:00|98223099999|9.3|15.0|10.5|11.6|15.9|16.6|
> > > |2023-07-26 18:00:00|59559099999|10.8|16.7|10.5|10.1|15.8|15.2|
> > > |2023-07-27 00:00:00|59559099999|7.2|13.0|9.8|10.6|13.3|14.5|
> > > |2023-07-27 06:00:00|59559099999|7.5|11.0|8.5|9.1|10.6|10.3|
> > > |2023-07-27 12:00:00|59358099999|2.0|5.0|2.3|3.8|5.8|6.3|
> > > |2023-07-27 18:00:00|59358099999|14.5|24.0|13.3|12.9|22.8|22.4|
> > > |2023-07-28 00:00:00|59134099999|5.5|10.0|5.0|4.6|11.0|11.5|
> > > |2023-07-28 06:00:00|59134099999|2.0|6.0|3.1|3.5|5.1|4.5|
> > > |2023-07-28 12:00:00|58921099999|1.7|2.7|1.5|1.9|2.4|3.2|
> > > |2023-07-28 18:00:00|58725099999|1.9|1.0|2.0|2.3|1.9|2.1|
> > > |2023-07-29 00:00:00|58506099999|1.6|1.0|2.5|2.7|1.7|2.3|
> > >
> > > As shown in the table, the results support your hypothesis. GraphCast, trained on ERA5 reanalysis, produces smoother wind forecasts that closely follow ERA5 and underestimate peak observations. In contrast, PhysicsFormer, trained on in-situ station data, better captures observed extremes at both 1 and 2 day lead times.

---

### Official Review · Reviewer_3psu · 2026-03-11

**Soundness:** 3
**Presentation:** 4
**Significance:** 3
**Originality:** 4
**Overall Recommendation:** 5
**Confidence:** 4

**Summary:**

In this paper, the authors aim to address the following limitations in station weather forecasting: 1) A diverse high-quality weather dataset spanning geographical locations all over the globe, 2) A strong baseline model tailored for the exact task of station weather forecasting and 3) Metric for evaluating the quality of prediction on extreme weather events.

They address the first point by introducing WEATHER-5K, a dataset composed of around 5600 stations spanning the globe from 2014 to 2023 at an hourly frequency with a clear training-validation-test split. This dataset differs from existing ones in that it contains recent observations and is much more diverse geographically.

The second point is addressed by introducing PhysicsFormer, a physics-informed transformer-based architecture that can take a sequence of stations as input and produce predictions at these stations for subsequent time-steps. It differs from the literature by incorporating a physics loss that makes sure the predictions are physically sound in addition to having a dynamics core that integrates simplified atmospheric dynamics.

For the third and last point, they introduce a new metric *Symmetric Extremal Dependence Index* (SEDI) which effectively counts the frequency at which both the predictions and the ground-truth "agree" on extreme values.

**Compliance With Llm Reviewing Policy:**

Affirmed.

**Ethical Review Flag:**

Flag this paper for an ethics review.

**Ethics Expertise Needed:**

["Research Integrity Issues (e.g., plagiarism)"]

**Final Justification:**

The rebuttal addressed my concerns. My main concern was originality (see weakness 1) but I believe it's up to the AC to decide on it so my score now reflects the contents of the submission and that only.

**Key Questions For Authors:**

* Figure 1: It would be better to have the scale and values in the y-axis for each variable. It's possible to flip the axis so that a larger area means smaller values for MAE.
* Temporal alignment: Aren't you concerned that linear interpolation might alter the ground-truth value? High-frequency components might get smoothed out because of this. Why not use ERA5 reanalysis as you already do for some stations?
* Dataset outlier detection: What's the exact process for detecting those and what do you do with them?
* For ERA5 reanalysis, how do you compute the missing values at particular stations? Do you use some kind of linear interpolation from the surrounding coordinates at which the data from ERA5 is available?
* I am confused about how exactly $X^{phys}_{t+1:t+\tau}$ is derived, could you provide more details?

**Limitations:**

Yes.

**Strengths And Weaknesses:**

**Strengths**:
- Their proposed physics-informed transformer-based architecture outperforms the machine-learning based baselines while being very competitive with the NWP one (even outperforming it on some variables).
**Weaknesses**:
* The WEATHER-5K dataset along with the SEDI metric was already introduced in (Han, et al 2024; arxiv: 2406.14399v1) so as far as I can tell, this is plagiarism as it copies the very table and figures themselves.
* Figure 6e shows the discrepancy between ERA5 and in-situ observations and it's not negligible, yet it's used as ground-truth in the proposed dataset notably for missing data which means that the models will be trained and evaluated on biased data. While the authors acknowledge this limitation, they provide no concrete evidence on its effects besides saying that its effects are "minimal".
* While the SEDI metric penalizes predictions that overestimate minimal values (or underestimate maximal ones), it doesn't do so for predictions that may underestimate minimal values further or when it overestimates maximal ones.
* The authors do not provide other metrics that aim to compare the frequency profile of predictions compared to the ground-truth (e.g. through the power spectra) nor do they evaluate the physical soundness of their predictions.
* No ablations are provided for the PhysicsFormer architecture, namely the role of the physics-based losses and the dynamic core
* No physics-informed approach is present in the baselines. For example, it will be interesting how climODE fares against PhysicsFormer

---

> ### Author Rebuttal · Authors · 2026-03-31
>
> # W1
> We sincerely thank the reviewer for their rigorous review. To strictly maintain the double-blind reviewing process, we cannot confirm or deny authorship of specific non-anonymous preprints. However, we respectfully note that the conference policy explicitly allows submissions to be posted as preprints on platforms like arXiv, and these preprints do not constitute prior publication or plagiarism. We firmly assure the reviewer that this submission is entirely our original work and that we have strictly adhered to all conference policies regarding anonymity and originality.
> # W2
> Please refer to reviewer ck3s W3 and W4.
> # W3
> SEDI evaluates whether predictions exceed extreme-event thresholds, without penalizing deviations beyond them, which suffices for real-world pre-warning. MSE and MAE (Table 2) capture prediction accuracy and deviation magnitude. Across 10 stations, differences between 99.9% and 99.5% thresholds are minor, indicating the 99.5% percentile reliably identifies extreme high-temperature events.
> |Station ID|Quantile(Temp)|
> |-|-|
> |40689099999|0.995=49.0,  0.999=50.7|
> |40833099999|0.995=49.0,  0.999=50.0|
> |40831099999|0.995=49.0,  0.999=50.2|
> |40811099999|0.995=48.7,  0.999=50.0|
> |40676099999|0.995=48.6,  0.999=50.0|
> |40680099999|0.995=48.4,  0.999=49.9|
> |40582099999|0.995=48.1,  0.999=50.0|
> |40670099999|0.995=48.0,  0.999=49.0|
> |40674099999|0.995=47.5,  0.999=48.8|
> |40794099999|0.995=47.5,  0.999=48.9|
> # W4
> We conducted frequency-based analysis to assess physical consistency. Below Table show our method achieves high frequency correlations across all variables, outperforming baselines. Full spectra plots are provided in https://521.im/RDGX4i6r.png.
> | Model| Temp   | Dewpoint | Wind Angle | Wind Rate | Sea-level Pressure |
> |-|-|-|-|-|-|
> | PatchTST     | 0.9043 | 0.3168   | 0.9404     | 0.8071    | 0.4674|
> | Informer| 0.9226 | 0.9819   | 0.3769 | 0.7259    | 0.7399|
> | iTransformer| 0.9704 | 0.9888   | 0.3866 | 0.8795    | 0.9231|
> | Pyraformer   | 0.8871 | 0.9841   | 0.4677 | 0.8683    | 0.7673|
> | **Ours** |**0.9611**|**0.9968**|**0.9322**|**0.9828**|**0.9709**|
> # W5
> Please refer to reviewck 3s
> # W6
> ClimODE is CNN-based and cannot be directly applied to our discrete, irregularly distributed weather stations, as it is designed to operate on regular grids. To enable a comparison, we supplement our experiments by generating ClimODE predictions on a regular grid and interpolating them to our station locations. This allows evaluation on the same station-level data as PhysicsFormer.
> |Method|MAE MSE(Temp)|MAE MSE(Dew)|MAE MSE(Wind)|MAE MSE(Sea Level Pressure)|
> |-|-|-|-|-|
> |climODE |3.35  15.20|3.15. 16.24|2.70  9.40|4.56  38.80|
> |**Ours**|**2.34  11.86**|**2.51  14.01**|**1.44  4.55**|**3.53  32.41**|
> # Q1
> We clarify that in Figure 1, MAE values for each variable are normalized across methods:
> $$ \tilde{\mathrm{MAE}}_v = \frac{\mathrm{MAE}_v - \min(\mathrm{MAE}_v)}{\max(\mathrm{MAE}_v) - \min(\mathrm{MAE}_v)} $$
> The plot is flipped so that larger areas correspond to lower MAE and thus better performance. Since the values are normalized, the y-axis has no absolute units.
> # Q2
> As described in the Section 3.2 (line 189), small timestamp deviations are handled by using the nearest observation within a 30-minute window, and linear interpolation over the surrounding 12 hours is applied only for a small fraction of remaining missing hours. While ERA5 reanalysis could be used for interpolation, it introduces a systematic discrepancy with true station observations—for example, the annual mean temperature difference is around 2-3°C (Figure 6e). For very short gaps, interpolating from nearby observations better preserves high-frequency local variations and results in lower error than ERA5-based interpolation. To systematically evaluate these approaches, we randomly masked a fraction of the complete station data and filled the missing values using three interpolation methods. The resulting errors are summarized in the table below:
> |Variable|Linear Interpolation Error (RMSE)|ERA5 Interpolation Error (RMSE)|Hybrid Interpolation Error (ours) (RMSE)|
> |-|-|-|-|
> |Temp| 2.67 | 2.34 | **2.15** |
> |Dew| 2.80 | 2.42 | **2.21** |
> |Wind| 2.44 | 2.08 | **1.86** |
> |Pressure| 2.02 | 1.68 | **1.47** |
> # Q3
> Outliers are identified by examining temporal dynamics and flagging observations that fall outside physically plausible ranges or deviate strongly from local trends. Specifically, values exceeding climatological limits or sudden deviations beyond 5 standard deviations of hourly differences are marked, and lightweight machine learning filters help distinguish genuine extremes from sensor errors or spurious noise. Detected outliers are then treated as missing and imputed via forward-fill or linear interpolation within each station, ensuring temporal continuity while preserving physically consistent atmospheric patterns for robust model training.
> # Q4 and Q5
> Yes, we use linear interpolation. For Q5 please refer to reviewer WpCq W1.

---

### Official Review · Reviewer_WpCq · 2026-03-13

**Soundness:** 4
**Presentation:** 4
**Significance:** 4
**Originality:** 3
**Overall Recommendation:** 5
**Confidence:** 5

**Summary:**

This paper introduces a novel spatiotemporal dataset, Weather-5K, designed to evaluate time-series models on weather station forecasting tasks. The authors provide a detailed comparison between Numerical Weather Prediction (NWP) and time-series forecasting methods, supported by extensive and robust experiments. Additionally, the paper proposes PhysicsFormer, a novel physics-embedded spatiotemporal forecasting method. Experimental results demonstrate that PhysicsFormer comprehensively outperforms classical time-series methods, achieving state-of-the-art (SOTA) performance that is second only to NWP.

**Compliance With Llm Reviewing Policy:**

Affirmed.

**Final Justification:**

The authors adequately addressed all questions and weaknesses raised in the review during the rebuttal period. I have raised the soundness score from 3 to 4, and I maintain a positive stance on this paper.

**Key Questions For Authors:**

Q1: In the second paragraph, is it appropriate to categorize NWP methods into "physically-based" and "data-driven" models? NWP conventionally refers strictly to numerical weather prediction methods that solve partial differential equations.

Q2: Given that the Integrated Surface Dataset (ISD) contains over 20,000 stations, what specific factors or patterns of missing observations led to the rigorous filtering down to only 5,000 stations?

**Limitations:**

See Weaknesses and Questions.

**Strengths And Weaknesses:**

Strengths:

S1: Introduces a high-quality spatiotemporal station dataset, advancing the time-series forecasting field from a data perspective.

S2: The paper is logically structured. The main figure effectively illustrates the proposed benchmarking methodology.

S3: Extensive experiments are conducted on the Weather-5K dataset, thoroughly demonstrating the performance of various time-series forecasting algorithms in the spatiotemporal domain.

S4: Innovatively proposes a novel physics-guided spatiotemporal sequence forecasting algorithm.

S5: Provides runnable code, ensuring the validity and reproducibility of the experimental results.

Weaknesses:

W1: Lacks sufficient technical details regarding PhysicsFormer, such as the specific design of the "dynamic core".

W2: The experimental section could be strengthened by incorporating comparisons with recent GNN-based and Large Time Series methods.

W3: Including 1-2 AI-based weather prediction models (e.g., GraphCast and Pangu-weather) as baselines would significantly bridge the gap between weather forecasting and time-series forecasting, thereby making the paper more convincing.

---

> ### Author Rebuttal · Authors · 2026-03-31
>
> # W1
> We sincerely thank the reviewer. We have expanded the "dynamic core" section in the revised manuscript (Appendix B). Below is the detailed formulation.
>
> Since our dataset comprises 5,672 highly discrete stations, directly computing spatial gradients (e.g., $\nabla P$) as in grid-based NWP models is infeasible. To bridge this, the **Dynamic Core** is designed as a graph-based neural ODE system that mimics simplified atmospheric dynamics.
>
> **1. Spatial Graph for Gradient Approximation:** We construct a spatial graph $\mathcal{G}$ connecting stations via K-Nearest Neighbors based on Haversine distance, enabling message-passing to approximate spatial derivatives (divergence/advection). The approximated gradient for variable $u$ is:
> $$ \nabla u_i \approx \sum_{j \in \mathcal{N}(i)} \frac{w_{ij}}{\sum w_{ij}} (u_j - u_i) W_{\nabla} $$
>
> **2. Governing Equations on Graph:** Let $X_{i,t} = [T, W, P, D]$ represent temperature, wind, pressure, and dew point. The dynamic core models the time derivative $\frac{d X_{i,t}}{d t}$ driven by physical forces (momentum by $\nabla P$, thermodynamics by wind advection, mass by wind divergence). We parameterize these coupled derivatives with a lightweight GNN:
> $$ \frac{d X_{i,t}}{d t} = \Phi_{\theta}(X_{i,t}, \nabla X_{i,t}) $$
>
> **3. Autoregressive Integration:** We use a 4th-order Runge-Kutta (RK4) integrator to bridge continuous dynamics with discrete forecasting. Starting from historical state $X_{t}$, we compute future states:
> $$ X_{t+k}^{phys} = X_{t+k-1}^{phys} + \int_{t+k-1}^{t+k} \Phi_{\theta}(X^{phys}(s), \mathcal{G}) ds \quad \text{for } k=1, 2, ..., \tau $$
>
> **4. Transformer Residual:** The final prediction combines the physically constrained base forecast $X_{t+1:t+\tau}^{phys}$ with a Transformer-based localized residual $\Delta X_{t+1:t+\tau}^{res}$ as in Eq. (5).
> This explains the effectiveness of the physical loss: the dynamic core provides a structured mass-momentum prior, while the Transformer handles non-linear localized anomalies.
> # W2
> We conducted additional experiments including comparisons with recent GNN-based and large-scale time series models. The results are summarized below.
> |Type|Method|MAE MSE(Temp)|MAE MSE(Dew)|MAE MSE(Wind)|MAE MSE(Sea Level Pressure)|
> |-|-|-|-|-|-|
> |GNN-based|T-PatchGNN(ICML2024)[1]|5.37  87.15|5.23 78.91|1.88 6.34|5.78 62.26|
> |LTMs|Time-MoE(ICLR2025)[2]|2.58  15.11|2.79  17.35|1.50  4.91|4.02  38.97|
> |Physics-based|**Ours**|**2.34  11.86**|**2.51  14.01**|**1.44  4.55**|**3.53  32.41**|
>
> [1] Zhang *et. al.*, Irregular multivariate time series forecasting: A transformable patching graph neural networks approach.
>
> [2] Shi *et. al.*, Time-moe: Billion-scale time series foundation models with mixture of experts.
>
> # W3
> We conducted additional experiments to compare with recent AI-based weather prediction models (e.g., GraphCast, Pangu-weather, and Fengwu).
>
> To obtain station-level predictions for these AI weather models, we downloaded their official checkpoints and ran inference using ERA5 to generate 6-hourly forecasts for a 7-day horizon. We then interpolated the grid prediction of the corresponding variables to the exact latitude and longitude coordinates of our 5,672 weather stations, and finally calculated the errors against the actual station observations. The results are summarized below:
>
> | Method | Temp (MAE/MSE) | Wind (MAE/MSE) | SLP (MAE/MSE) |
> |---|---|---|---|
> | ECMWF-HRES | 1.9 / 8.6 | 1.6 / 5.0 | 1.3 / 5.2 |
> | GraphCast | 2.1 / 9.8 | 1.7 / 5.6 | 1.9 / 11.3 |
> | Pangu-weather| 2.2 / 10.2| 1.8 / 6.2 | 2.3 / 15.9|
> | Fengwu | 2.0 / 9.4 | 1.7 / 5.6 | 2.1 / 16.3|
>
> # Q1
> We agree that Numerical Weather Prediction (NWP), in its conventional definition, specifically refers to physically-based methods that solve governing atmospheric partial differential equations. Our original phrasing was therefore imprecise in grouping both physically-based and learning-based approaches under the umbrella of NWP. To address this, we will revise the manuscript to clearly distinguish between physically-based NWP methods and data-driven weather forecasting models.
>
> # Q2
> The reported 20,000+ ISD stations correspond to the cumulative number of stations with observations from 1901–present, rather than the number of stations active in any given year. In practice, station availability varies significantly over time. As noted in Sec. 3.1, restricting to stations with at least 10 years of **continuous operation** reduces this to 10,701 candidates. Among these, many exhibit substantial missing data, often in the form of long gaps in hourly records. To ensure data quality and temporal consistency, we further retain only stations with more than 90% valid hourly observations, yielding 5,672 stations (Sec. 3.2). Thus, the reduction is mainly due to temporal sparsity and stringent filtering of stations with significant missing or discontinuous observations.

---

> > ### Author Rebuttal · Reviewer_WpCq · 2026-04-04
> >
> > The authors have provided the requested details on the dynamic core and added more baselines. The new benchmark and physics-guided method are significant contributions to time series forecasting. Evaluating models on a **real-world weather dataset** effectively fills a current gap in the field. Additionally, the station-based physics-guided method (distinct from grid-based data) is challenging and constitutes a solid contribution.

---

> > > ### Author Response · Authors · 2026-04-04
> > >
> > > We sincerely thank the reviewer for the encouraging feedback and for recognizing the core value of our work. We are deeply grateful that you appreciate both of our main contributions: the introduction of the real-world Weather-5K benchmark to evaluate existing TSF models, and the challenging design of our station-based, physics-guided weather prediction algorithm. Your recognition that this benchmark and methodology effectively fill a crucial gap in the field, pushing TSF models toward practical weather applications, means a lot to us. We will incorporate all these valuable discussions and newly added baselines into the final manuscript. **Thank you for your strong support!**

---

### Decision · Program_Chairs · 2026-04-30

**Decision:**

Accept (regular)

**Comment:**

This paper proposes WEATHER-5K, a large-scale global weather station dataset, and PhysicsFormer, a physics-informed spatiotemporal forecasting model. Its main strengths include the high-quality benchmark dataset advancing real-world time-series forecasting evaluation, PhysicsFormer's competitive performance against operational NWP systems, and its novel graph-based dynamic core combined with Transformer residuals. After the rebuttal, concerns related to missing technical details of the dynamic core, absent GNN/AI weather model baselines, missing ablations, ERA5 data leakage risk, and plagiarism allegations have been largely resolved.